

# Updated spectral radiance calibration on TIR bands for the TANSO-FTS-2 onboard GOSAT-2

Hiroshi Suto[1], Fumie Kataoka[2], Robert O. Knuteson[3], Kei Shiomi[1], Nobuhiro Kikuchi[1], Akihiko Kuze[1]

[1]Japan Aerospace Exploration Agency, Tsukuba-city, Ibaraki, 305-8505, Japan
5  [2]Remote Sensing Technology Center of Japan, Tsukuba-city, Ibaraki, 305-8505, Japan
[3]University of Wisconsin-Madison, Madison, WI, 53706, USA

*Correspondence to*: Hiroshi Suto (suto.hiroshi@jaxa.jp)

10  **Abstract.** The Thermal and Near-Infrared Sensor for Carbon Observation Fourier-Transform Spectrometer-2 (TANSO-FTS-2) onboard the Japanese Greenhouse gases Observing SATellite-2 (GOSAT-2) observes a wide spectral region of the atmosphere, from the ShortWave-InfraRed (SWIR) to longwave Thermal InfraRed radiation (TIR) with 0.2cm$^{-1}$ spectral intervals. The TANSO-FTS-2 has operated nominally since Feb 2019, and the atmospheric radiance spectra it has acquired have been released to the public. This paper describes an updated model for spectral radiance calibration and its validation. 15  The model applies to the version 210210 TIR products of the TANSO-FTS-2 and integrates polarization sensitivity correction for the internal optics and the scanner mirror thermal emission. These correction parameters are characterized by an optimization which depends on the difference between the spectral radiance of the TANSO-FTS-2 and coincident nadir observation data from the Infrared Atmospheric Sounding Interferometer (IASI) on METOP-B. To validate the updated spectral radiance product against other satellite products, temporally and spatially coincident observation points were 20  considered for Simultaneous Nadir Overpass (SNO) from February 2019 to March 2021 from the Atmospheric Infrared Sounder (AIRS) on Aqua, IASI on METOP-B, and TANSO-FTS on GOSAT. The agreement of brightness temperatures between the TANSO-FTS-2 and AIRS and IASI was better than 0.3 K (1$\sigma$) from 180 K to 330 K for the 680 cm$^{-1}$ $CO_2$ channel. The brightness temperatures between the TANSO-FTS-2 and TANSO-FTS of version v230231, which implemented a new polarization reflectivity of the pointing mirror and was released in June 2021, generally agree from 220 25  K to 320 K. However, there is a discrepancy at lower brightness temperatures, pronounced for $CO_2$ channels at high latitudes. To characterize the spectral radiance bias for along-track and cross-track angles, a 2-Orthogonal Simultaneous Off-Nadir Overpass (2O-SONO) is now done for the TANSO-FTS-2 and IASI, the TANSO-FTS-2 and AIRS, and the TANSO-FTS-2 and TANSO-FTS. The 2O-SONO comparison results indicate that the TIR product for the TANSO-FTS-2 has a bias that exceeds 0.5 K in the $CO_2$ channel for scenes with forward and backward viewing angles greater than 20°. These multi- 30  satellite sensor and multi-angle comparison results suggest that the calibration of spectral radiance for the TANSO-FTS-2 TIR, version v210210, is superior to that of the previous version in its consistency of multi-satellite sensor data. In addition, the paper identifies the remaining challenging issues in current TIR products.



## 1 Introduction

Greenhouse gases Observing SATellite-2 (GOSAT-2), launched on 29 October 2018, extending the success of the Greenhouse gases Observing SATellite (GOSAT) (Kuze et al., 2009, 2012, 2016) mission. It carried the Thermal And Near infrared Sensor for carbon Observation Fourier-Transform Spectrometer-2 (TANSO-FTS-2) (Suto et al., 2021). To provide continuous monitoring of the global distribution of $X_{CO2}$ and $X_{CH4}$, GOSAT-2 measures both the ShortWave InfraRed (SWIR) solar radiation reflected from the earth's surface and the Thermal InfraRed (TIR) radiation from the ground and the

atmosphere. GOSAT-2 has extended SWIR spectral coverage beyond GOSAT capabilities. One extension is toward the shortwave for solar-induced fluorescence; another is toward the longwave for carbon monoxide (CO) in the 2.3 um region. Also, TIR spectral coverage is divided into two regions, band 4 (5.5 - 8.6 um) and band 5 (8.6 - 14.3 um). In addition, simultaneous spectral radiance observation of SWIR and TIR supports retrieving the lower tropospheric $CO_2$ and $CH_4$ concentrations. It leads to new applications for local emission estimation (Kuze et al., 2022).

Characterization of these spectral radiance is essential to provide consistent spectral radiance products for greenhouse-gas-observing satellites such as GOSAT, Orbiting Carbon Observatory-2 (OCO-2) in orbit since July 2014 (Crisp et al., 2004, 2008, 2017), Orbiting Carbon Observatory-3 (OCO-3) in orbit since May 2019 (Eldering et al., 2019), the Sentinel-5 Precursor/TROPOspheric Monitoring Instrument (TROPOMI) in orbit since October 2017 (S5P) (Hu et al. 2018), and also the TIR sounders such as Infrared Atmospheric Sounding Interferometer (IASI) on METOP-B (Clebaux et al., 2009) and

Atmospheric Infrared Sounder (AIRS) on Aqua (Aumann et al., 2003). During GOSAT-2's first year of operation, several calibration processes for characterizing the TANSO-FTS-2 were carried out with onboard calibrators, as reported in Suto et al., 2021. In the early stage of the TANSO-FTS-2 calibration, we found a challenging issue with the TIR products, a brightness temperature bias for lower scene temperatures.

To reduce this bias, we reassessed the calibration model for the TIR bands of the TANSO-FTS-2. The new calibration

model and optimized calibration coefficients were derived by comparing well-characterized sensor data from other satellites. In addition, we showed that the spectral radiance for the TANSO-FTS-2 TIR bands is consistent with these satellites' inter-calibration data, with time-series and wavenumber dependencies.

This paper first introduces an updated instrument calibration model for the TANSO-FTS-2 TIR bands. A description of the optimization procedure follows for calibration coefficients, such as non-linear response, polarization sensitivity, scanner

mirror reflection, and scanner mirror's thermal emission. Next is a validation of updated radiance data with the first two years of in-orbit performance compared to temporally and spatially coincident data for Simultaneous Nadir Overpasses (SNOs) from other satellites. Furthermore, these data were acquired for cross-track, along-track 2-Orthogonal Simultaneous Off-Nadir Overpass (2O-SONO) data from other satellites to validate multi-angle consistency.



## 2 Instrument calibration models

All the processing from interferogram to atmospheric radiance spectra for the TANSO-FTS-2 was performed on the ground. The basic procedure is described in the GOSAT-2 Level-1 Algorithm Theoretical Basis Document (GOSAT-2 FTS-2 L1 ATBD, 2020) and Suto et al., 2021. As described in the previous paper, version v102102 of the TIR product has applied an empirical bias correction coefficient to reduce the brightness temperature bias for the TANSO-FTS-2 product. However, that product still has a low brightness temperature bias for cold scenes against the other coincident satellite data comparisons. To

update the physical model for correcting the low brightness temperature bias, a non-linear response, polarization sensitivity of internal optics, and thermal emission from scanner mirror are reassessed in this paper.

### 2.1 Non-linear correction

In level 1 processing, the raw digital signals are converted into physical units. For the TANSO-FTS-2, an interferogram was

constructed with a DC offset and gain correction. The simplified equation for conversion from raw digital units to physical units is described by equation (1).

$$I_{amp,b} = \frac{ADC\_scale_b}{PGA_{gain_b}} \cdot DN_b \cdot + DAC_{scale_b} \cdot DC_{offset_b} + V_{offset,b}$$

(1)


where

$b$:                 Bands (bands 4, 5)

$I_{amp,b}^X$:          Interferogram with DC offset and gain correction applied.

$ADC\_scale_b$:     Analog-to-digital conversion scale

$DN_{b,d}$:           Digital number for each interferogram

$PGA_{gain_b}$:      Gain factor for each band

$DAC\_scale_b$:     Digital-to-analog conversion factor for each band

$DC\_offset_{b,d}$:   DC offset clamped at start of observation

$V_{offset,b}$:         Offset signal


If the detector electronic chains have a non-linear response, the non-linear correction is applied in the interferogram domain as conventional signal processing. Equation (2) expresses the non-linear signal correction with quadratic and cubic terms. Here, $a_{nlc,b}$, $b_{nlc,b}$ and $c_{nlc,b}$ are non-linear coefficients for the quadratic factor, cubic factor, and offset, respectively.



$$I_{nlc,b} = I_{amp,b} - a_{nlc,b} \cdot I_{amp,b}{}^2 - b_{nlc,b} \cdot I_{amp,b}{}^3 + c_{nlc,b}$$

(2)

A Photo Conductive – Mercury Cadmium Telluride (PC-MCT) detector has a non-linear response with a quadratic term. The following model considers up to the linear and quadratic terms.

Nominally, interferogram signals have both AC and DC components. Then, the interferogram signals for each band (b) can be described with $AC_b$ and $DC_b$ components, as shown by equation (3).

$$I_{amp,b} = AC_b + DC_b$$

(3)

In this case, equation (2) with a quadratic term is rewritten as equation (4)

$$I_{nlc,b} = -a_{nlc,b}AC_b^2 + \left(1 - 2a_{nlc,b}DC_b\right)AC_b + \left(DC_b - a_{nlc,b}DC_b^2\right)$$

(4)

As a result of the fast-Fourier transform, equation (4) is converted to equation (5)

$$fft(I_{nlc,b}) = -(a_{nlc,b})(S_b \otimes S_b) + \left(1 - 2a_{nlc,b}DC_b\right)S_b$$

(5)

where
$S_b = fft(AC_b)$

$fft$  : Fast-Fourier transform operator

$\otimes$: Convolution operator

In the spectral domain, the $S_b$ component contains the in-band signal whereas the $S_b \otimes S_b$ component is the second
harmonic which is mainly outside the in-band region but in principle could overlap the edges of the in-band signal. Figure 1 shows the $S_b$ and $S_b \otimes S_b$ signals in the spectral domain for both TANSO-FTS and TANSO-FTS-2.  Both TANSO-FTS and TANSO-FTS-2 have a wideband TIR channel; however, the TIR channel of the TANSO-FTS-2 is separated into two bands regions. As shown in Fig. 1, $S_b \otimes S_b$ components (blue lines in Fig. 1.) overlap in the in-band signal (black lines) region for TANSO-FTS band 4, and it is prohibitively difficult to remove these components. In contrast, the $S_b \otimes S_b$ component is fully
separated in the TANSO-FTS-2 bands 4 and 5, and these components are negligible in the spectral domain. The signal in the spectral domain is expressed as equation (6).





$$fft(I_{nlc,b}) \sim (1 - 2a_{nlc,b}DC_b)S_b$$

(6)


This equation suggests that a non-linear correction can be applied in the spectral domain with only the non-linear coefficient $a_{nlc,b}$ , the $DC_b$ component, and the in-band spectrum $S_b$.

## 2.2 Polarization correction model

In a previous paper (Suto et al., 2021), we reported the low brightness temperature bias in TIR bands 4 and 5 for the version v102102 product. To correct this bias, we implemented a polarization sensitivity correction for the TANSO-FTS-2 because the internal optical components are based on the high-polarization-sensitivity materials, such as ZnSe. To account for the polarization sensitivity correction for the version v210210 level 1 algorithm, the calibration equations are modified from those of version v102102.

The detailed polarization sensitivity of the TANSO-FTS-2 optics is modeled by Stokes vectors and Mueller matrices, as expressed in the optical efficiency of the FTS mechanism and aft-optics, phase difference due to the pointing mirror reflectivity, and CT rotation, respectively ( $M_{opt}$, $M_r$, $M_\varepsilon$ and $M_{mirror}$ are Muller matrices of two orthogonal polarization beam splitters). $S_{T\_output}$ , $S_{T\_input}$ , $S_{T\_mirror}$ are output and input signals for Stokes vector. In this case, the $S_{T\_output}$ is expressed as equation (7).


$$S_{T\_output} = M_{opt}M_r(-\theta_{CT})M_{mirror}M_r(\theta_{CT})S_{T\_input} + M_{opt}M_r(-\theta_{CT})M_\varepsilon M_r(\theta_{CT})S_{T\_mirror} + S_{Backgroud}$$

(7)

where


$$S_{T\_input} = \begin{bmatrix} B(T_{scene}) \\ 0 \\ 0 \\ 0 \end{bmatrix}$$

$$M_{opt} = \frac{1}{2}\begin{bmatrix} p_2^2(\sigma) + q_2^2(\sigma) & p_2^2(\sigma) - q_2^2(\sigma) & 0 & 0 \\ p_2^2(\sigma) - q_2^2(\sigma) & p_2^2(\sigma) + q_2^2(\sigma) & 0 & 0 \\ 0 & 0 & 2p_2(\sigma)q_2(\sigma) & 0 \\ 0 & 0 & 0 & 2p_2(\sigma)q_2(\sigma) \end{bmatrix}$$






$$M_{mirror} = \frac{1}{2}\begin{bmatrix} p_1^2(\sigma) + q_1^2(\sigma) & p_1^2(\sigma) - q_1^2(\sigma) & 0 & 0 \\ p_1^2(\sigma) - q_1^2(\sigma) & p_1^2(\sigma) + q_1^2(\sigma) & 0 & 0 \\ 0 & 0 & 2p_1(\sigma)q_1(\sigma) & 0 \\ 0 & 0 & 0 & 2p_1(\sigma)q_1(\sigma) \end{bmatrix}$$

$$M_r(\theta_{CT}) = \begin{bmatrix} 1 & 0 & 0 & 0 \\ 0 & cos2\theta_{CT} & -sin2\theta_{CT} & 0 \\ 0 & sin2\theta_{CT} & cos2\theta_{CT} & 0 \\ 0 & 0 & 0 & 1 \end{bmatrix}$$


$$M_\varepsilon = E - M_{mirror} = E - \frac{1}{2}\begin{bmatrix} p_1^2(\sigma) + q_1^2(\sigma) & p_1^2(\sigma) - q_1^2(\sigma) & 0 & 0 \\ p_1^2(\sigma) - q_1^2(\sigma) & p_1^2(\sigma) + q_1^2(\sigma) & 0 & 0 \\ 0 & 0 & 2p_1(\sigma)q_1(\sigma) & 0 \\ 0 & 0 & 0 & 2p_1(\sigma)q_1(\sigma) \end{bmatrix}$$

Then,

$$S_{obs} - S_{ds} = \frac{B(T_{scene})}{4}\left( \left(p_2^2(\sigma) + q_2^2(\sigma)\right)\left(p_1^2(\sigma) + q_1^2(\sigma)\right) + \left(p_2^2(\sigma) - q_2^2(\sigma)\right)\left(p_1^2(\sigma) - q_1^2(\sigma)\right) \right)$$

$$- \frac{B(T_{mirror})}{2}\left(p_2^2(\sigma) - q_2^2(\sigma)\right)\left(p_1^2(\sigma) - q_1^2(\sigma)\right)$$

(8)

$$S_{bb} - S_{ds} = \frac{B(T_{bb})}{4}\left( \left(p_2^2(\sigma) + q_2^2(\sigma)\right)\left(p_1^2(\sigma) + q_1^2(\sigma)\right) - \left(p_2^2(\sigma) - q_2^2(\sigma)\right)\left(p_1^2(\sigma) - q_1^2(\sigma)\right) \right)$$

(9)


To derive the $B(T_{scene})=L_{b,d}^{obs}$ , finally, equation (10) is obtained.

$$L_{b,d}^{obs} = \left[\frac{S_{b,d}^{obs} - S_{b,d}^{ds}}{S_{b,d}^{ict} - S_{b,d}^{ds}}\right] \cdot \left[\frac{\left(p_2^2(\sigma) + q_2^2(\sigma)\right)\left(p_1^2(\sigma) + q_1^2(\sigma)\right) - \left(p_2^2(\sigma) - q_2^2(\sigma)\right)\left(p_1^2(\sigma) - q_1^2(\sigma)\right)}{\left(p_2^2(\sigma) + q_2^2(\sigma)\right)\left(p_1^2(\sigma) + q_1^2(\sigma)\right) + \left(p_2^2(\sigma) - q_2^2(\sigma)\right)\left(p_1^2(\sigma) - q_1^2(\sigma)\right)}\right] B_{b,d}^{ict}$$

$$+ \left[\frac{2\left(p_2^2(\sigma) - q_2^2(\sigma)\right)\left(p_1^2(\sigma) - q_1^2(\sigma)\right)}{\left(p_2^2(\sigma) + q_2^2(\sigma)\right)\left(p_1^2(\sigma) + q_1^2(\sigma)\right) + \left(p_2^2(\sigma) - q_2^2(\sigma)\right)\left(p_1^2(\sigma) - q_1^2(\sigma)\right)}\right] L_{b,d}^{m\_obs}$$

(10)

The term in equation (11) already corrected the non-linear effects.





$$\left[\frac{S_{b,d}^{obs} - S_{b,d}^{ds}}{S_{b,d}^{ict} - S_{b,d}^{ds}}\right]$$

180 (11)

So, if we consider the non-linear effect based on equation (6), the equation (11) is extracted as equation (12).

$$\left[\frac{\left(1 - 2a_2 p_g DC_{obs}\right) \cdot S_{b,d}^{obs} - \left(1 - 2a_2 DC_{ds}\right) \cdot S_{b,d}^{ds}}{\left(1 - 2a_2 DC_{ict}\right) \cdot S_{b,d}^{ict} - \left(1 - 2a_2 DC_{ds}\right) \cdot S_{b,d}^{ds}}\right] = \left[\frac{\frac{\left(1 - 2a_2 p_g DC_{obs}\right)}{\left(1 - 2a_2 DC_{ds}\right)} \cdot S_{b,d}^{obs} - S_{b,d}^{ds}}{\frac{\left(1 - 2a_2 DC_{ict}\right)}{\left(1 - 2a_2 DC_{ds}\right)} \cdot S_{b,d}^{ict} - S_{b,d}^{ds}}\right]$$

185 (12)

where $p_g$ is the polarization sensitivity gain for non-linearity of nadir observation signal against calibration signal for $DC_{obs}$. $DC_b$ is independently observed and related to the cross-track angle. During both blackbody and deep-space calibration, the polarization axis of the internal optics is rotated at 90° from the nadir observation. The polarization sensitivities between

calibration and nadir observation show gains due to the difference in input optical angles.

Finally,

$$L_{b,d}^{obs} = \left[\frac{\frac{\left(1 - 2a_2 p_g DC_{obs}\right)}{\left(1 - 2a_2 DC_{ds}\right)} \cdot S_{b,d}^{obs} - S_{b,d}^{ds}}{\frac{\left(1 - 2a_2 DC_{ict}\right)}{\left(1 - 2a_2 DC_{ds}\right)} \cdot S_{b,d}^{ict} - S_{b,d}^{ds}}\right] \cdot \left[\frac{\left(p_2^2(\sigma) + q_2^2(\sigma)\right)\left(p_1^2(\sigma) + q_1^2(\sigma)\right) - \left(p_2^2(\sigma) - q_2^2(\sigma)\right)\left(p_1^2(\sigma) - q_1^2(\sigma)\right)}{\left(p_2^2(\sigma) + q_2^2(\sigma)\right)\left(p_1^2(\sigma) + q_1^2(\sigma)\right) + \left(p_2^2(\sigma) - q_2^2(\sigma)\right)\left(p_1^2(\sigma) - q_1^2(\sigma)\right)}\right] B_{b,d}^{ict}$$

$$+ \left[\frac{2\left(p_2^2(\sigma) - q_2^2(\sigma)\right)\left(p_1^2(\sigma) - q_1^2(\sigma)\right)}{\left(p_2^2(\sigma) + q_2^2(\sigma)\right)\left(p_1^2(\sigma) + q_1^2(\sigma)\right) + \left(p_2^2(\sigma) - q_2^2(\sigma)\right)\left(p_1^2(\sigma) - q_1^2(\sigma)\right)}\right] L_{b,d}^{m\_obs}$$

(13)

$$DC_{obs,ds,ict} = DAC_{scale} \cdot DC_{clamp\ for\ obs,ds,ict} + DC_{offset}$$

200 (14)

The spectral radiance seen by the TANSO-FTS-2 instrument when viewing the black body (ict, or internal calibration target) is a contamination of a direct emission from the blackbody and reflected radiance originating from various external surfaces that the black body views. The viewing factor for each component is expressed as follows:






$$B_b^{ict}[n] = C_b^{ict}[n] + C_b^{ict\_baffle}[n] + C_b^{SAA\_str}[n] + C_b^{OMA}[n] + C_b^{BS}[n]$$

(15)

$$C_b^{ict}[n] = \varepsilon_b^{ict} \cdot L_b\left(\sigma_b[n], T^{ict}\right)$$

210 (16)

$$C_b^{ict\_baffle}[n] = \left(1 - \varepsilon_b^{ict}\right) \cdot \varepsilon_b^{ict\_baffle} \cdot A^{ict\_baffle} \cdot L_b\left(\sigma_b[n], T^{SSA+Y}\right)$$

(17)

$$C_b^{SAA\_str}[n] = \left(1 - \varepsilon_b^{ict}\right) \cdot \varepsilon_b^{SAA\_str} \cdot A^{SAA\_str} \cdot L_b\left(\sigma_b[n], T^{SSA-Y}\right)$$

(18)

$$C_b^{OMA}[n] = \left(1 - \varepsilon_b^{ict}\right) \cdot \left(1 - \varepsilon_b^{scanner\_mirror}\right) \cdot \varepsilon_b^{OMA} \cdot \left(A^{OMA}\right) \cdot L_b\left(\sigma_b[n], T^{IOA+Z}\right)$$

(19)

$$C_b^{BS}[n] = \left(1 - \varepsilon_b^{ict}\right) \cdot \left(1 - \varepsilon_b^{scanner\_mirror}\right) \cdot A^{BS} \cdot L_b\left(\sigma_b[n], T^{BS}\right)$$

(20)

$$A^{ict\_baffle} + A^{SAA\_str} + A^{OMA} + A^{BS} = 1$$

220 (21)

where

| | |
|---|---|
| $p_1^2(\sigma), q_1^2(\sigma):$ | Scanner reflectance for p and s |
| $p_2^2(\sigma), q_2^2(\sigma):$ | Transmittance for p- and s-polarization signals for internal optics. |
| $L_b\left(\sigma_b[n], T^{ict}\right):$ | Radiance for temperature $T^{ict}$, and wavenumber $\sigma_b[n]$ |
| $p_g:$ | Polarization sensitivity gain between calibration angles (ICT and Deep-space) and nadir observation |
| $\varepsilon_b^{ict\_baffle}:$ | ICT baffle surface emissivity in band b |
| $A^{ict\_baffle}:$ | ICT view of ICT baffle |
| $\varepsilon_b^{SAA\_str}:$ | SAA surface emissivity in band b |
| $A^{SAA\_str}:$ | ict view of SAA structure |
| $\varepsilon_b^{OMA}:$ | OMA surface emissivity in band b |
| $A^{OMA}:$ | ict view of OMA structure |
| $A^{BS}:$ | ict view of BS |
| $\varepsilon_b^{scanner\_mirror}:$ | Scan mirror surface emissivity |






### 2.3 Mirror reflectance model

Due to the large mirror size, it is difficult to measure the mirror reflectance onboard the TANSO-FTS-2 instrument directly. During prelaunch calibration, the complex index of the mirror sample was characterized simultaneously with that of the actual mirror. Consequently, the scan mirror reflectance is expressed as the following equations with the complex spectral

index of refraction of the mirror coating $m$.

$$cos\theta_i = \frac{cos(CT_{ang}) \cdot sin(AT_{ang}) + cos(AT_{ang})}{\sqrt{2}}$$

(22)


$$r_p(m, \theta_i) = \frac{m^2 cos\theta_i - \sqrt{m^2 - sin^2\theta_i}}{m^2 cos\theta_i + \sqrt{m^2 - sin^2\theta_i}}$$

(23)

$$r_s(m, \theta_i) = \frac{cos\theta_i - \sqrt{m^2 - sin^2\theta_i}}{cos\theta_i + \sqrt{m^2 - sin^2\theta_i}}$$

(24)


$$p_1^2(\sigma) = r_p(m, \theta_i) \cdot r_p^*(m, \theta_i) = \frac{\left|m^2 cos\theta_i - \sqrt{m^2 - sin^2\theta_i}\right|^2}{\left|m^2 cos\theta_i + \sqrt{m^2 - sin^2\theta_i}\right|^2}$$

(25)

$$q_1^2(\sigma) = r_s(m, \theta_i) \cdot r_s^*(m, \theta_i) = \frac{\left|cos\theta_i - \sqrt{m^2 - sin^2\theta_i}\right|^2}{\left|cos\theta_i + \sqrt{m^2 - sin^2\theta_i}\right|^2}$$

255 (26)

The emissivity of the scan mirror is expressed in equation (27).

$$\varepsilon_b^{scanner\_mirror}(\sigma) = 1 - \frac{1}{2}[p_1^2(\sigma) + q_1^2(\sigma)]$$


(27)



## 3 Optimization of instrument models

The calibration equation and related models are described in the previous section. The calibration procedure must be optimized for maximum spectral radiance accuracy. In this section, the optimization procedure for the above models is
discussed.

Usually, the non-linear effect of a low-temperature scene is smaller than that of high-temperature scene. We obtained the non-linear quadratic coefficient with a high-temperature target in the interferogram domain during the prelaunch calibration test. A non-linear coefficient is determined which minimizes the out-of-band signal intensity of low-frequency components.

The first term of equation (10) is the main part of the polarization effect. We assume that the difference in spectral radiance in selected spectral regions between the TANSO-FTS-2 and the coincident dataset, especially at low temperatures, is directly related to polarization correction terms. We derive the ratio of p and s internal optics against wavenumber based on the IASI matchup dataset. This step makes use of the value of mirror reflectance obtained during the prelaunch test where the initial parameters for polarization sensitivities are determined.

In the next step, the polarization sensitivity is further optimized with a non-linearity correction based on equation (13). In this optimization, we changed the domain from interferogram to spectra to reduce the unknown parameters with the spectra domain. As expressed in equations (1) and (2), a total of five parameters (ADC conversion scale, gain factor, DAC conversion scale, offset signal, and non-linearity correction coefficients) have to be considered to derive a precise interferogram. In contrast, in the spectral domain, the parameters are non-linear correction coefficients and DC offset as
expressed in equation (12) except for polarization sensitivity gain. Then, the polarization sensitivity, non-linear correction coefficients, DC offset, and polarization sensitivity gain are optimized with equation (13) to minimize the difference of spectral radiance between the TANSO-FTS-2 and IASI with SNO condition. The variation range of brightness temperature between the TANSO-FTS-2 and IASI is wider than that of AIRS, then the SNO condition for IASI data is applied.

The optimized results of polarization sensitivity are presented in Fig. 2. This value is applied in version 210210
products with prelaunch scan mirror reflectance.

## 4 Inter-comparisons with reference satellite sensors

The comparison of the TANSO-FTS-2 TIR band nadir and off-nadir comparisons provide a quantitative spectral assessment of the radiometric bias relative to the AIRS on AQUA, IASI on METOP-B, and TANSO-FTS on GOSAT.

In the following section, two types of coincident criteria are applied: SNO and cross-track, along-track 2O-SONO.
Conventional weather satellites sensors, such as AIRS and IASI, have only observation capability in cross-track motion because the scanning motion is only performed in cross-track. In contrast, the TANSO-FTS-2 and TANSO-FTS accommodate a two-axis agile pointing system to target the interesting observation location. Then, the TANSO-FTS-2 can coordinate the cross-track of the TANSO-FTS-2 and the cross-track of other satellites, and the along-track of the TANSO-



FTS-2 and cross-track of other satellites. The schematic diagrams of 2O-SONO coincident observation images are illustrated in Fig. 3. The coincidence criteria for SNO and 2O-SONO with satellite sensors are listed in Table 1. The coincident latitudes between AIRS and the TANSO-FTS-2, between IASI and the TANSO-FTS-2, and between TANSO-FTS and the TANSO-FTS-2 are illustrated in both SNO (a) and 2O-SONO (b) in Fig. 4. The coincident points between the AIRS and the TANSO-FTS-2 are in the mid-latitudes, and those of IASI and the TANSO-FTS-2 are located at high latitudes. In contrast, the coincident points between TANSO-FTS and the TASNO-FTS-2 cover the complete range of latitudes pole-to-pole. These leads to a comparison with different brightness temperature ranges for each matching dataset. We focused on the comparison in the following spectral regions: $CO_2$ channel (681.99 - 691.66 $cm^{-1}$), window channel (900.3 - 903.78 $cm^{-1}$), $O_3$ channel (1030.08 - 1039.69 $cm^{-1}$), and $CH_4$ channel (1304.36 - 1306.68 $cm^{-1}$) same as previous our estimation (Suto et al., 2021).

As for AIRS data, AIRS L1C data were applied (AIRS Science Team/Strow 2019). For the IASI, IASI-B data were selected from the NOAA CLASS archive. To compare TANSO-FTS and the TANSO-FTS-2, version 230231 of TANOS-FTS, released on June 2021, was selected. This version has improved the consistency between AIRS and IASI for a better polarization coefficient of the pointing mirror.

## 4.1 Comparison between AIRS and the TANSO-FTS-2, IASI and the TANSO-FTS-2, and TANSO-FTS and the TANSO-FTS-2 with SNO condition

Figure 5 shows the brightness temperature differences (the TANSO-FTS-2 values minus other satellite values) in 1 K grided bin average at four focused channels against the window temperature between the TANSO-FTS-2 of version v210210, AIRS, IASI, and TANSO-FTS for SNO. The brightness temperature difference between the TANSO-FTS-2 of version v102102 and AIRS, IASI, and TANSO-FTS are also plotted in Fig. 5 for reference. The data periods for each comparison are listed in Table 2. Figure 5 suggests that version v210210 products are more consistent with AIRS and IASI data than version v102102 in all channels, especially in low-temperature window. In addition, the low-temperature biases and significant deviations were removed in version v210210 products in the channels of $CH_4$. The statistical analysis results are also summarized in Table 2. As suggested in Table 2, the deviation (SD) between the TANSO-FTS-2 and AIRS, IASI is reduced with version v210210, especially in channels for $CO_2$ and $CH_4$. In comparing the TANSO-FTS-2 and TANSO-FTS, the deviation is increased with version v210210. As shown in Fig. 5, in the temperature range from 180 K to 240 K, TANSO-FTS product presents large positive values against the TANSO-FTS-2 for $CO_2$ and $CH_4$ channels. This means that the TANSO-FTS has inconsistent values at lower temperatures, especially for $CO_2$ and $CH_4$. In addition, the negative values are detected from 240 to 260 K in the $CH_4$ channel. The previous version of the TANSO-FTS-2 has negative biases at low temperatures. The consistency between the TANSO-FTS-2 and TANSO-FTS agrees in these regions. In other words, version v210210 of the TANSO-FTS-2 products removes the low-temperature biases, even though TANSO-FTS version v230231 still has lower temperature biases.



Figures 6 presents the time series of the brightness temperatures difference between the TANSO-FTS-2 and IASI, between the TANSO-FTS-2 and AIRS, and between the TANSO-FTS-2 and TANSO-FTS for four channels, both versions of v210210 and v102102. During winter in the southern hemisphere, version v102102 products present negative values and
large deviations due to seasonal variation, especially in the $CO_2$ and $CH_4$ channels. Cold temperature scenes over Antarctica were selected as coincident observation locations. In contrast, the version v210210 products suggest no seasonal variation except for the TANSO-FTS comparison. These plots also indicate that version v230231 of TANSO-FTS products has a negative bias against cold scenes, observed over high-latitude coincident points.

As a result of SNO, version v210210 of the TANSO-FTS-2 products show the averaged bias is less than +/-0.3 K for all
four channels. In addition, the deviations against IASI and AIRS for the $CO_2$ and $CH_4$ channels are less than 0.3 K and 0.5 K, respectively. These results suggest that the consistency for the $CO_2$ and $CH_4$ channels between the TANSO-FTS-2 and AIRS, between the TANSO-FTS-2 and IASI, are much improved. The comparison between the TANSO-FTS-2 and TANSO-FTS has a discrepancy in the low-temperature region, but we concluded that version v230231 of TANSO-FTS product has a challenging issue at low temperatures, especially at high latitudes, for both $CO_2$ and $CH_4$ channels. Therefore, the calibration
of the TIR band for TANSO-FTS will be updated in the next version of the level 1 product to improve the consistency of brightness temperature, especially in low-temperature high-latitude regions.

**4.2 Comparison between AIRS and the TANSO-FTS-2, IASI and the TANSO-FTS-2, and TANSO-FTS and the TANSO-FTS-2 with 2O-SONO condition**

As described in the previous section, version v210210 of the TANSO-FTS-2 product agrees with AIRS and IASI products in nadir coincident observations. In the next step, the comparison on 2O-SONO was made to confirm the incident angle dependency of the TANSO-FTS-2 observations. The coincident conditions for 2O-SONO are listed in Table 1.

Figure 7 presents the brightness temperatures difference between the TANSO-FTS-2 and AIRS, the TANSO-FTS-2 and IASI, the TANSO-FTS-2 and TANSO-FTS with the TANSO-FTS-2 1° bin averaged along and cross-track angles. The
deviation of each bin is plotted with shaded lines. The coincident observations between the TANSO-FTS-2 and the AIRS were selected +40° and -40° of the AIRS cross-track angle and the related along-track of the TANSO-FTS-2 listed in Table 1. For the IASI cross-track angle, +20° and -20° of the IASI data were selected.

Figure 7(a) shows that the brightness temperature difference between the TANSO-FTS-2 and AIRS is almost stable within the +/-10° along-track angle. Figure 7 (a) also suggests that the brightness temperature difference depends on the
track angle of the TANSO-FTS-2 over a +/-10° along-track angle. As shown in Fig. 7 (a), the brightness temperature difference between the TANSO-FTS-2 and AIRS increased for larger along-track angles of the TANSO-FTS-2. In contrast, the dependence of cross-track angle plotted in Fig. 7(b) is not clear except for the $CH_4$ channel in a cross-track angle of 5° to 10°.





Figures 7 (c) and (e) also present the brightness temperature difference between the TANSO-FTS-2 and IASI and

between the TANSO-FTS-2 and TANSO-FTS against an along-track angle for the TANSO-FTS-2, respectively. These plots also suggest that the brightness temperature difference depends on the along-track angle of the TANSO-FTS-2. The dependence is almost flat between -10° to +10° of the along-track angle. This is a similar feature to the results of the AIRS comparison. Figures 7 (d) and (f) show the brightness temperature difference between the TANSO-FTS-2 and IASI and between the TANSO-FTS-2 and TANSO-FTS against a cross-track angle for the TANSO-FTS-2, respectively. Figure 7 (d)

suggests that the brightness temperature difference does not depend on the cross-track angle of the TANSO-FTS-2 in the channels of $CO_2$, $CH_4$, $O_3$, and window. In contrast, a cross-track dependency is observed for the $CH_4$ and $O_3$ channels in Fig. 7 (f), which compares the TANSO-FTS-2 and TANSO-FTS.

Figure 8 shows that 1° along-track by 1° cross-track grid average brightness temperature difference between the TANSO-FTS-2 and the AIRS, between the TANSO-FTS-2 and IASI, and between the TANSO-FTS-2 and TANSO-FTS.

These figures also clearly present the dependence on the along-track angle, especially in the $CO_2$ channel. For the TANSO-FTS comparison, a cross-track angle dependence is also observed, even though the results of comparing AIRS and IASI are not supported. Comparing Figs. 7 (f) and 8 (f), we found that the brightness temperature difference with the significant cross-track angle condition shows large biases.

As presented in Fig. 5, TANSO-FTS has a lower temperature bias in the $CO_2$ and $CH_4$ channels in a SNO. Therefore,

the brightness temperature differences at four channels in the 1K gridded average against the window temperature are plotted in Fig. 9 for 2O-SONOs. As shown in Fig. 9, the lower temperature bias in TANSO-FTS is the same as SNO. In addition, a high-temperature bias in the $CH_4$ channel is the same as the TANSO-FTS. Therefore, we conclude that the TANSO-FTS-2 does not have a cross-track dependence on TANSO-FTS. The feature is related to the brightness temperature bias on TANSO-FTS version v230231 products.

Compared with TANSO-FTS, this difference may indicate a scan angle dependence of the scene selection mirror, which is not entirely removed by the polarization correction performed in the processing v230231. The available along-track range of TANSO-FTS is +/-20°. In contrast, the TANSO-FTS-2 can be set between +/-40°. In this comparison, the matchups are selected between -10° and +10° of the TANSO-FTS-2 along-track angle.

As presented in Fig. 9, the agreement between the TANSO-FTS-2, AIRS, and IASI is well. However, the agreement

between the TANSO-FTS-2 and TANSO-FTS is worse than the comparison against AIRS and IASI. This suggests that the calibrated radiance of TANSO-FTS, especially in low brightness temperature regions, still has a small bias. A summary of the inter-comparisons between the TANSO-FTS-2 and multi-satellite sensors with SONO is listed in Table.3.



## 5 Conclusions

This paper reports the performance of the TANSO-FTS-2 bands 4 and 5 with the new radiance calibration method. The method is based on a non-linear response, a polarization sensitivity correction in internal optics, and scanner mirror thermal emission in the spectral domain. To evaluate its performance, the spectral radiances (level 1 processor version v210210) collected by the TANSO-FTS-2 between February 2019 and October 2021 are compared to both the simultaneous nadir and 2-orthogoanl off-nadir observations of the AIRS on AQUA, IASI on METOP-B, TANSO-FTS on GOSAT for the TIR bands.

We conclude that the agreement between the TANSO-FTS-2 and AIRS, IASI is better than 0.3 K for scenes brighter than 220 K in the $CO_2$ and $CH_4$ channels. Compared with AIRS and IASI, TANSO-FTS has a small bias on the brightness temperature for low temperatures. In the latest version of v230231 for TANSO-FTS, the polarization correction parameter for the pointing mirror is improved and officially released. In the nominal temperature region, such as 280 K brightness temperature, the agreement between the TANSO-FTS-2 and TANSO-FTS is well. However, only TANSO-FTS suggests the

brightness temperature bias in a cold scene over a high latitude region and is a challenge of TANSO-FTS. In addition, the result of 2O-SONO indicates that the TANO-FTS-2 has an along-track angle depending on bias over +/-10° along-track angle. The agreement between the TANSO-FTS-2 and AIRS/IASI is good for the nominal pointing angle. However, for forward- or backward-viewing with a pointing angle greater than 20° the estimated bias exceeds 0.5 K bias in the $CO_2$ channel for TANSO-FTS-2 version 210210.

*Data availability.*

All datasets used here are publicly available and can be accessed through the links and references provided.

*Author contributions.*

HS wrote the manuscript and analyzed data with support from FK, RO, and KS. RO, FK, KS, NK, and AK contributed to
interpreting the results. FK, RO, and KS supported to the satellite inter-comparison data preparation or expertise on data sets. All authors discussed the results and contributed to the manuscript.

*Competing interests.*

The authors declare that they have no conflict of interest.






*Acknowledgments.*

The authors would like to thank Y. Yata and H. Ochi of the Mitsubishi Space Software Corporation and the members of the Japanese Ministry of the Environment, the National Institute for Environmental Studies, L3 Harris, and ABB Inc. for their cooperation.


*Financial support.*

This work was funded by JAXA.



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




**Tables:**

**Table 1.** Temporally and spatially co-incident conditions for comparing SNO and 2O-SONO

| Coincident type | Satellite sensor | Distance between two orbits [km] | Time difference [min] | CT angle for TANSO-FTS-2 [deg.] | AT angle for TANSO-FTS-2 [deg.] | Distance between obs. Location [km] | AIRS Scan angle [deg.] | IASI scan angle [deg.] | TANSO-FTS scan angle [deg.] |
|---|---|---|---|---|---|---|---|---|---|
| SNO | AIRS | <+/- 100 | <+/-5 | <+/-3 | <+/-3 | <17 | - | - | - |
| | IASI | <+/- 100 | <+/-5 | <+/-3 | <+/-3 | <17 | - | - | - |
| | TANSO-FTS | <+/- 100 | <+/-5 | <+/-3 | <+/-3 | <17 | - | - | - |
| 2O-SONO | AIRS | <+/- 100 | <+/-30 | <+/-40 | <+/-35 | - | <+/-40 | - | - |
| | IASI | <+/- 100 | <+/-30 | <+/-40 | <+/-35 | - | - | <+/-20 | - |
| | TANSO-FTS | <+/- 100 | <+/-30 | <+/-40 | <+/-35 | - | - | - | <+/-15 AT <+/-35 CT |




**Table 2.** The averaged difference (Ave.) and deviation (SD.) of brightness temperatures between the TANSO-FTS-2 and multi-satellite sensors with SNO

| | No. of Matchups | Version | Period | $CO_2$ channel [K] | | Window channel [K] | | Ozone channel [K] | | $CH_4$ channel [K] | |
|---|---|---|---|---|---|---|---|---|---|---|---|
| | | | | Ave. | SD | Ave. | SD | Ave. | SD | Ave. | SD |
| AIRS-SNO | 573 | 102102 | Feb. 2019-Oct. 2020 | 0.01 | 0.21 | -0.63 | 2.55 | -0.45 | 1.55 | -0.11 | 1.16 |
| IASI-SNO | 1199 | 102102 | Feb. 2019-Mar. 2021 | -0.19 | 0.4 | -0.16 | 2.78 | -0.43 | 1.37 | -0.53 | 1.52 |
| TANSO-FTS -SNO | 72 | 102102 | Feb. 2019-Aug. 2020 | 0.16 | 0.28 | 0.0008 | 0.86 | -0.19 | 0.49 | -0.28 | 0.57 |
| AIRS-SNO | 573 | 210210 | Feb. 2019-Oct. 2020 | 0.15 | 0.18 | -0.17 | 2.59 | -0.01 | 1.56 | 0.11 | 0.41 |
| IASI-SNO | 1199 | 210210 | Feb. 2019-Mar. 2021 | -0.1 | 0.26 | -0.26 | 2.75 | -0.17 | 1.3 | 0.009 | 0.47 |
| TANSO-FTS -SNO | 72 | 210210 | Feb. 2019-Aug. 2020 | 0.3 | 0.35 | -0.06 | 0.85 | 0.07 | 0.53 | -0.13 | 0.74 |






**Table 3.** The averaged difference (Ave.) and deviation (SD.) of brightness temperatures between the TANSO-FTS-2 and

490                                    multi satellite sensors with 2O-SONO conditions

| | NO. of Matchups | Period | CO$_2$ channel [K] | | Window channel [K] | | Ozone channel [K] | | CH$_4$ channel [K] | |
|---|---|---|---|---|---|---|---|---|---|---|
| | | | Ave. | SD | Ave. | SD | Ave. | SD | Ave. | SD |
| AIRS-2O-SONO | 4062 | Feb. 2019-June. 2021 | 0.20 | 0.25 | 0.03 | 1.34 | -0.22 | 1.27 | -0.52 | 1.01 |
| IASI-2O-SONO | 6886 | Feb. 2019-Jul. 2021 | -0.05 | 0.26 | -0.10 | 1.71 | -0.08 | 0.81 | -0.04 | 0.90 |
| TANSO-FTS-2O-SONO | 116689 | Feb. 2019-Oct. 2021 | 0.12 | 0.41 | -0.17 | 1.13 | -0.13 | 0.78 | -0.51 | 1.05 |





**Figures:**

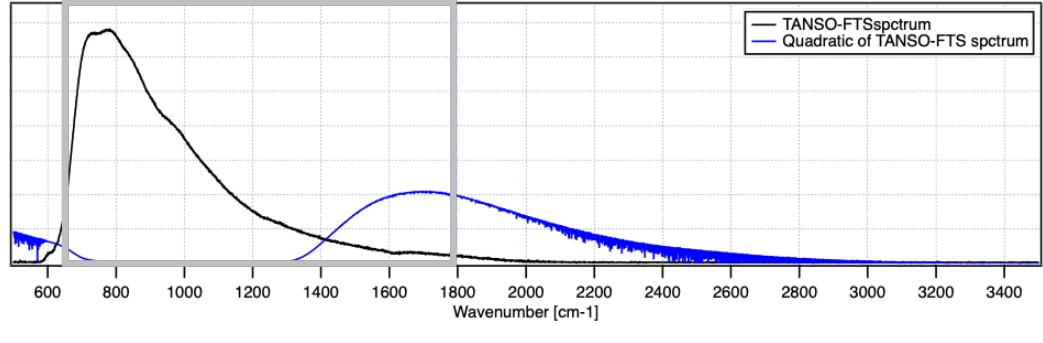

(a) TANSO-FTS

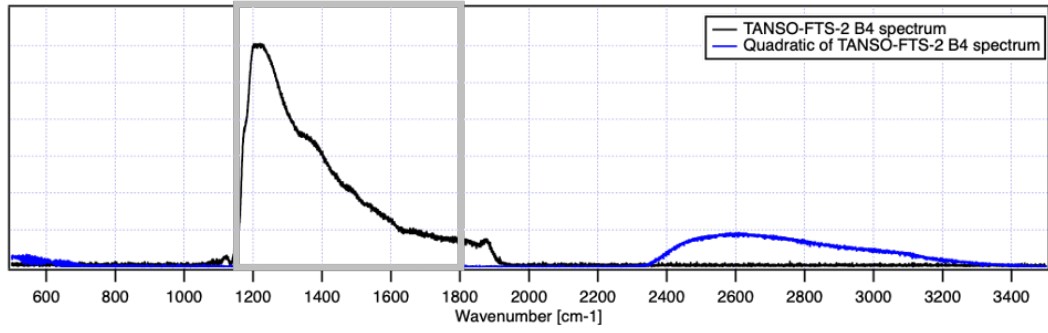

(b) Band 4 of TANSO-FTS-2

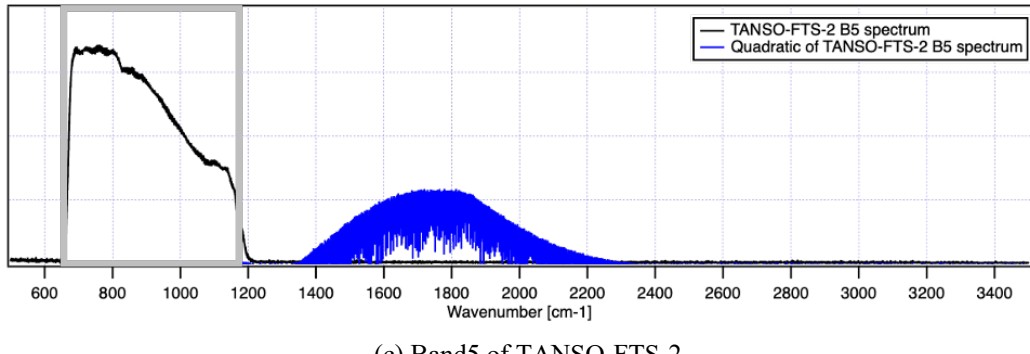

(c) Band5 of TANSO-FTS-2

Figure 1: Non-linear signals on the spectral domain for TANSO-FTS and TANSO-FTS-2.
Black lines present the original spectra. Blue lines show $S_b \otimes S_b$ components as the non-linear quadratic term after removing the original spectra.




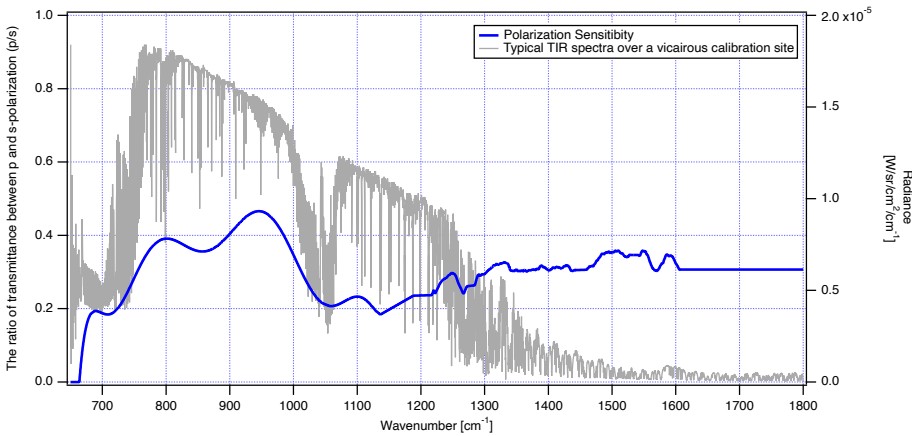


Figure 2: Polarization sensitivity model for bands 4 and 5. The blue line shows the polarization sensitivity as the transmittance ratio between p- and s-polarization against wavenumber. The gray line shows the observed spectral radiance in the TIR band for the TANSO-FTS-2.






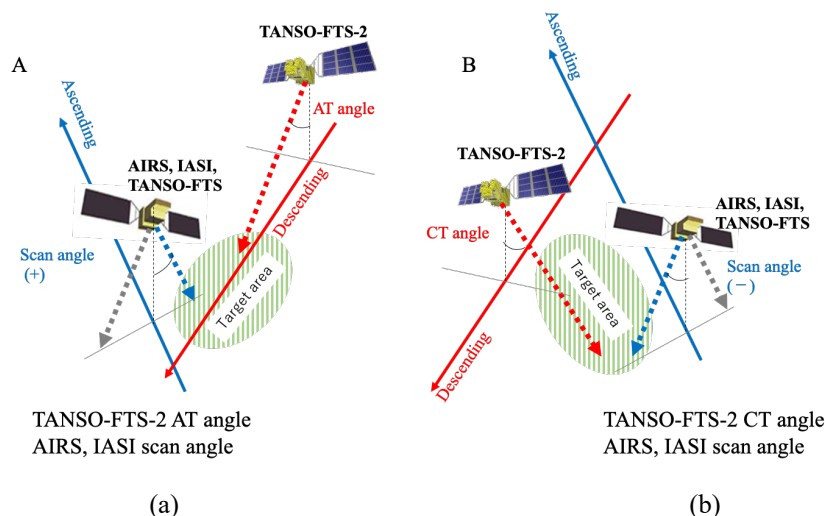

(a) (b)

Figure 3: The schematic diagram for coincident observation between the TANSO-FTS-2 and other satellites. (a) the
comparison between along-track observation by TANSO-FTS-2 and cross-track observation by other satellites (new method),
(b) the comparison between cross-track observation by the TANSO-FTS-2 and cross-track observation by other satellite
(conventional method).


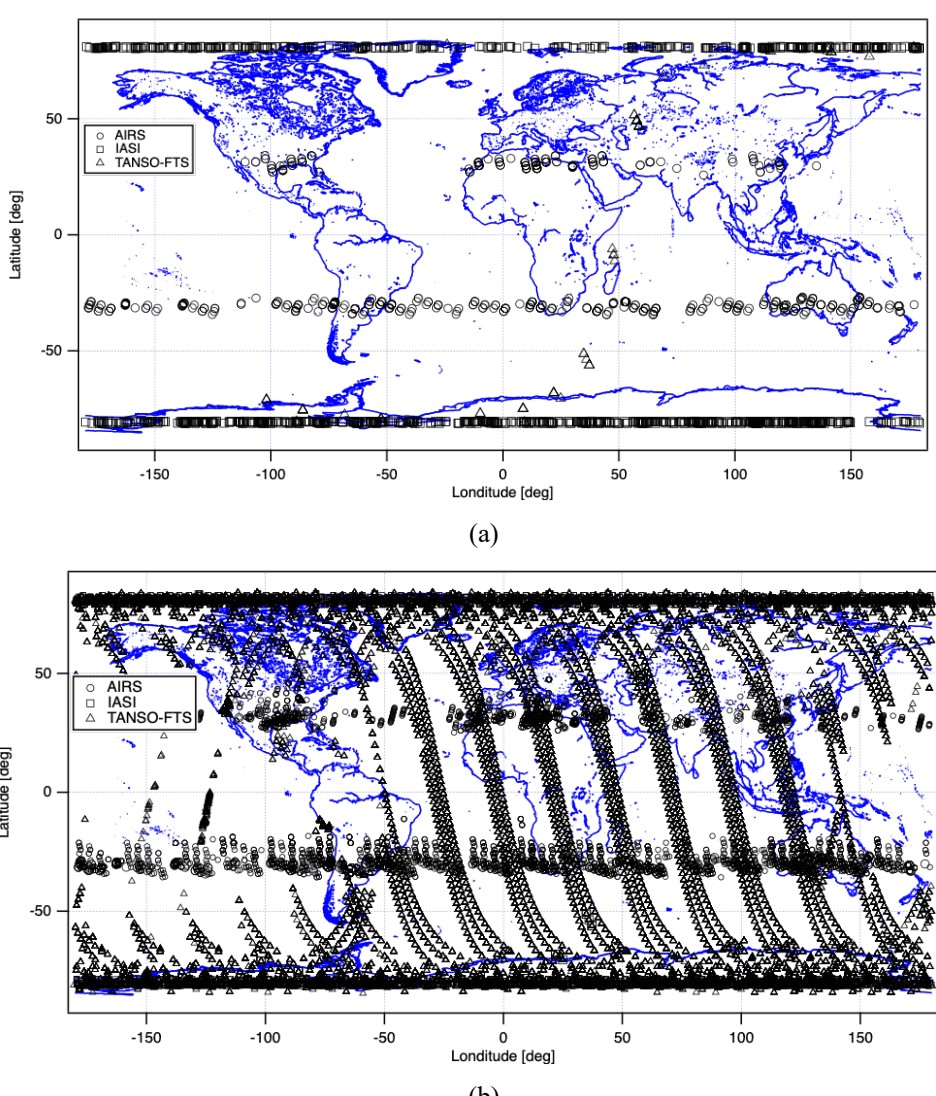

(a)

(b)


Figure 4: Comparing the TANSO-FTS-2 with other satellites: coincident latitude and longitude map between the TANSO-FTS-2 and AIRS/IASI/TANSO-FTS for SNO (a) and SONO(b).





(a)

(b)

(b)

(d)


Figure 5: The channel-dependent brightness temperature difference in 1 K gridded against window temperature for SNO condition between the TANSO-FTS-2 and AIRS/IASI/TANSO-FTS. (a) $CO_2$ channel, (b) $CH_4$ channel, (c) $O_3$ channel, (d) window channel. The filled dots are the data points, and each shade presents a standard deviation ($1\sigma$) for each 1 K grid.




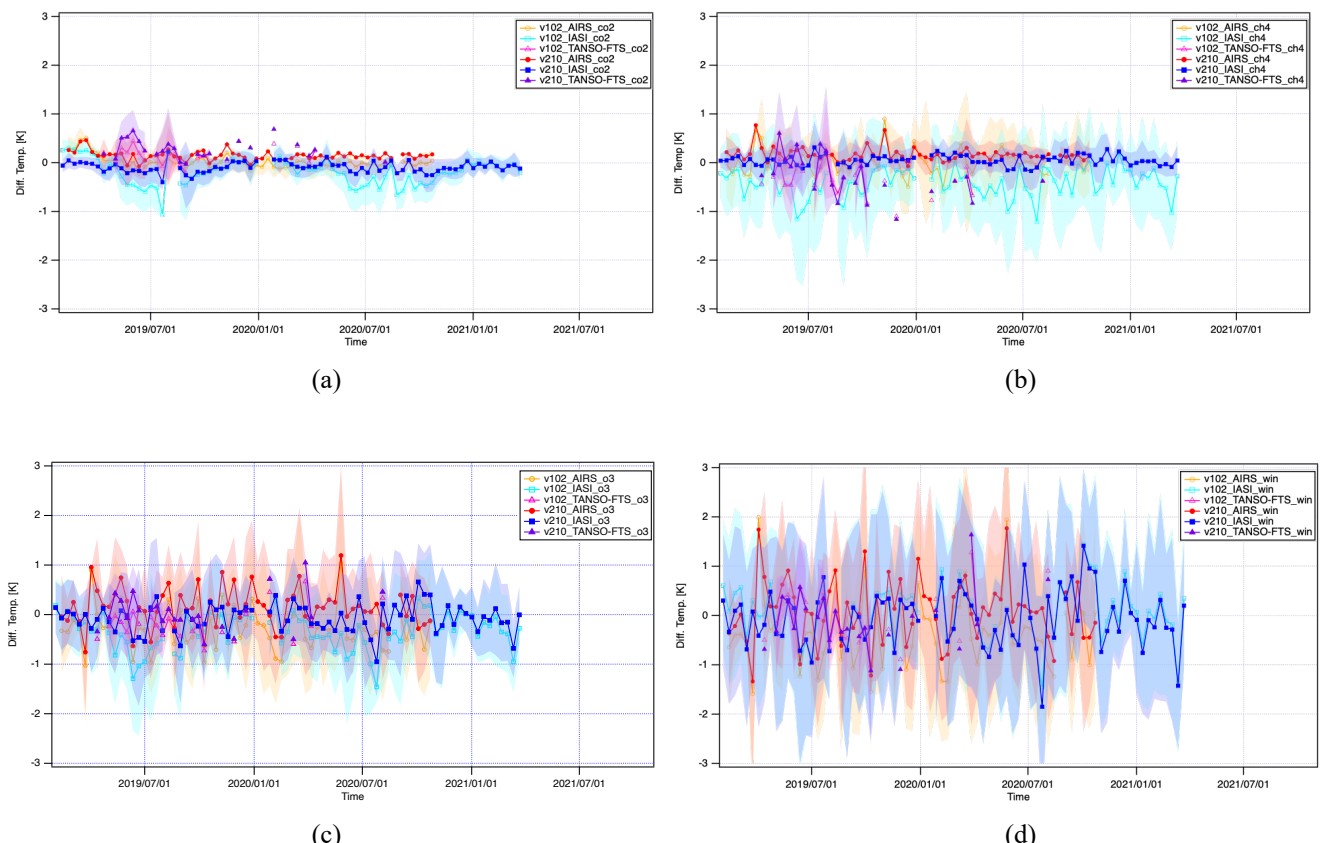

(a)                                                  (b)

(c)                                                  (d)


Figure 6: The channel-dependent brightness temperature difference for a ten-day average against window temperature for SNO condition between the TANSO-FTS-2 and AIRS/IASI/TANSO-FTS. (a) $CO_2$, (b) $CH_4$, (c) $O_3$, (d) window channels.





Figure 7: The channel dependent brightness temperature difference in 1° grided bin average against TANSO-FTS-2 AT
angle (left) and CT angle (right) for 2O-SONO for AIRS, IASI and TANSO-FTS. The shaded lines present the deviation
(1σ) for each grid. The gray bars indicate the number of averaged data in each bin.

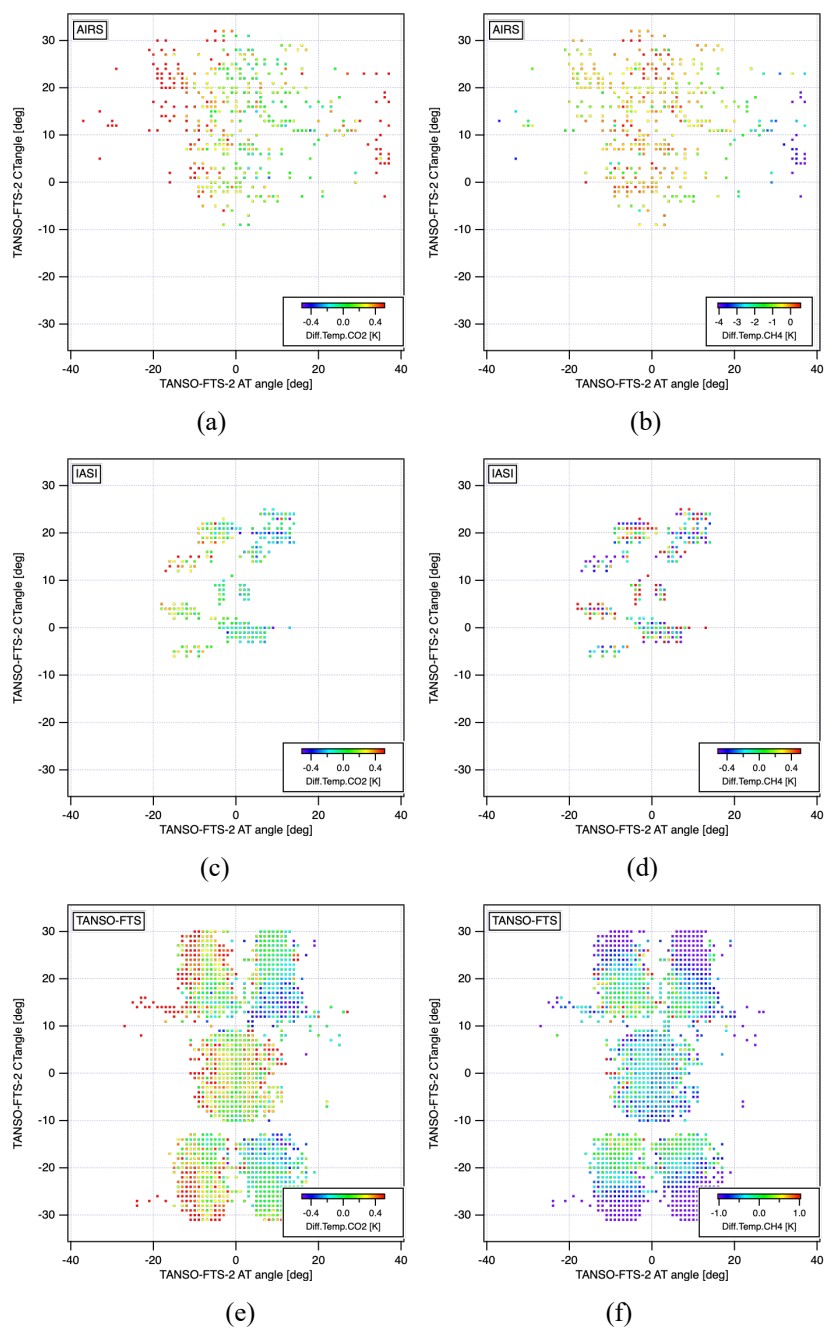



Figure 8: The 1°x1°gridded brightness temperature difference between the TANSO-FTS-2 and AIRS/IASI/TANSO-FTS for the $CO_2$ and $CH_4$ channels.




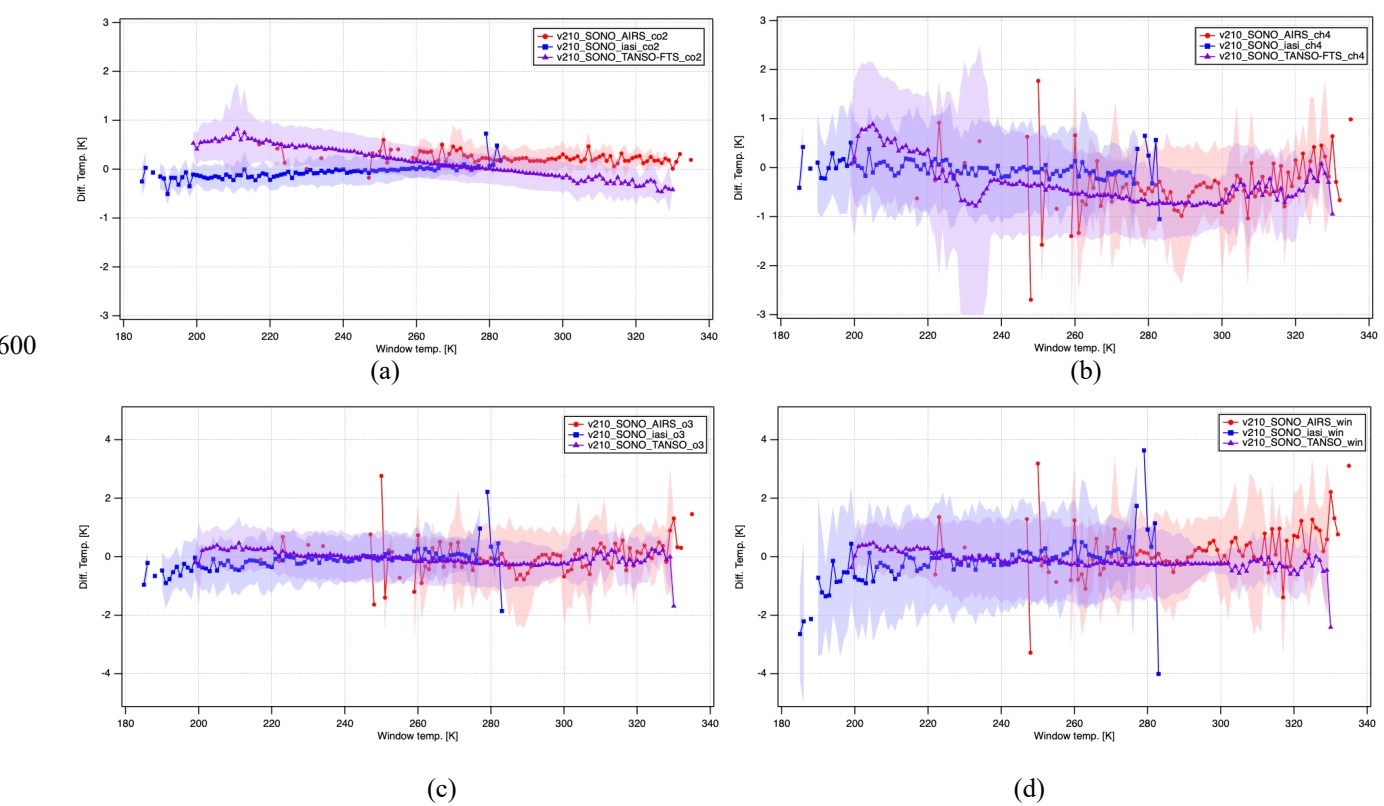

(a)

(b)

(c)

(d)

Figure 9: The channel-dependent brightness temperature difference in 1 K gridded average against window temperature for 2O-SONO between the TANSO-FTS-2 and AIRS/IASI/TANSO-FTS with deviation (shaded lines). (a) $CO_2$, (b) $CH_4$, (c) $O_3$, (d) window channels.
