# Peer review of "Updated spectral radiance calibration on TIR bands for TANSO-FTS-2 onboard GOSAT-2"

_Atmospheric Measurement Techniques, 2022_

## Referee Comment (RC2)

**Review of the AMT manuscript amt-2022-129**

**Overall assessment**

This is an important paper from the Japanese team of GOSAT (with one US co-author) on the radiometric calibration of TANSO-FTS-2 in the TIR bands with some discussion of the earlier TANSO-FTS instrument. The paper is rather technical but deserves publishing in AMT after accounting for the modifications suggested in this review and consideration of some questions. Figures, although numerous, are presenting in a compact and visual manner the main results of the radiometric differences between the Japanese instruments and two well validated infrared sounders (IASI from CNES and AIRS from JPL). An extensive reference is done to a previous paper on the subject by Suto et al. (2021). But as suggested below, the authors should help the reader to read this manuscript in a more stand-alone mode. A final recommendation is given at the end of this review.

Note that remarks from an anonymous reader at [https://doi.org/10.5194/amt-2022-129-RC1](https://doi.org/10.5194/amt-2022-129-RC1) are endorsed by the present review and should also be taken into account.

**Suggested detailed modifications**

The reviewer is using **l.xxx** (in bold) for the line to be modified and the convention old text → new text for the proposed changes. [A text between brackets is a comment/question/explanation as this is done here!]

General comment: one could just use TANSO-FTS-2 and TANSO-FTS instead of "the TANSO-FTS-2" and "the TANSO-FTS". This has been done in the following proposed modifications but not everywhere in the text.

**l.12** …to longwave Thermal InfraRed radiation (TIR) with 0.2cm$^{-1}$ spectral intervals.

→ …to the longwave Thermal InfraRed (TIR) with a 0.2 cm$^{-1}$ spectral sampling.

[Space between value and unit; "intervals" is a little confusing; one could add that the FWHM of the unapodized spectra is ~0.240 cm$^{-1}$ (in the infrared channels)]

**l.56** …consistent with these satellite's intercalibration data,…

→ …consistent with the intercalibration data of the other TIR sounders mentioned above,…

**l.70** …a non-linear response… → …the non-linear response of the infrared detectors…

**l.99** …considers up to the linear and quadratic terms.

→ …considers only the linear and quadratic terms (neglecting the cubic one).

**l.105** with a quadric term… → with a quadratic term only…

**L.140/l.142** Choose between Mueller or Muller but do not use both forms! What about Müller?]

**l.142** …and CT rotation… → …and cross-track (CT) rotation angle (called $\theta_{CT}$ in the following)…

[In several instances specific notations for the various parameters or variables are used in the equations but not defined early enough for the reader to follow them smoothly. The reviewer has tried to improve this situation. But the question of equations is also discussed below in a dedicated section of this review]

**l.176** The term in equation (11) already corrected the non-linear effects.

→ The multiplicative factor of the first term of equation (10) is called $Cal_b$ in the following equation (11) and includes the non-linearity correction. [Check if it is OK and see the discussion below on the equations concerning the subscripts $b$ and $d$]

l.179    Equation (11) should read $Cal_b$ = […]

l.181    …equation (6), the equation (11) is extracted as equation (12).

→ …equation (6), equation (11) can be recast as equation (12).

l.186    [The symbol $a_2$ is not specifically defined in equation (12). It should (perhaps) be related to $a_{nlc,b}$ of equation (2). The variable $p_g$ is not precisely defined. Overall the text there is hard to follow!]

[Check if equation (12) coul read:]        $Cal_b$ = […] = […]

l.189    the polarization axis of the internal optics is rotated at 90° from the nadir observation.

[This is not quite clear and the reviewer is suggesting the following]

→ the pointing mirror is rotated along its axis by ±90° (from the nadir observation) to view the deep space (ds) or the internal calibration target (ict).

[This would be consistent and will recall the notations introduced earlier in the text. The polarization effect is implicit because the incidence angle on the mirror would change. Relating the angle to the CT angle $\theta_{CT}$ already introduced would be even better. Could one specify that $\theta_{CT}$ = 0 is exact nadir?]

l.190    …the difference in input optical angles. → …the difference in incidence angle on the pointing mirror.

l.202    …(ict or internal target) is a contamination of a direct emission from the blackbody and reflected…

→ …(ict for internal target in the various symbols used) is a combination of the direct emission from the black body (at the temperature $T^{ict}$) and the reflected…

[One has to decide between black body and blackbody consistently in the text]

l.223    for p and s → for p and s polarizations (see subscripts p and s in the corresponding symbols)

l.224    Transmittance for p- and s-polarization signals for internal optics → Transmittance for the p- and s-polarized beams within the FTS

l.225    Radiance for temperature $T^{ict}$, and wavenumber $\sigma_b[n]$

→ Radiance for black body at a temperature $T^{ict}$ and wavenumber $\sigma_b[n]$

[The authors should better explain what is [$n$]. Is it the channel index ($CO_2$, window, $O_3$, $CH_4$) used in Tables 1, 2 and 3 as well as in the figures?]

l.226    …between calibration angles (ICT and Deep-space)

→ …between different pointing mirror angles towards the black body (ict in the various symbols) or deep space (ds in the various symbols)

l.229/230        [The symbol SAA is not precisely defined and could induce confusion with the South Atlantic Anomaly. Could the authors propose a different acronym?]

l.231/232        [Same question for the definition of OMA]

**l.238**   …complex index of the mirror sample… → …complex index of refraction of the mirror material (with coating)….

**l.239**   …actual mirror. → …actual flight mirror.

**l.240** [One could add:]

→ A star as superscript is used for the complex conjugate in equations (25) and (26).

**l.241**   Equation (22)

[The authors should define more precisely CTang and ATang and make the link with $\theta_{CT}$ already introduced or could possibly use $\theta_{AT}$? The angles CTang and ATang are appearing in Figure 3 and are called "AT angle" and "CT angle". This is fine but this is introducing new notations. TANSO-FTS is on the same footing as IASI and AIRS and the corresponding viewing direction across track is called "scan angle". A better explanation of these various angles (in particular a more consistent choice of notations) would be useful. The same type of comment is appearing several times below]

**L.259**   **[**Is there a difference between the "pointing mirror" and the "scan mirror"? The scan mirror could be understood as the moving mirror of the interferometer. This is to be clarified and consistent notations should be used in equations (27) and in **l.234**]

**l.272**   …the ratio of the p and s internal optics against… → …the ratio of the p and s transmission against…

[It is unclear at this point if it is the ratio of the signals at the output of the detector or if it is the ratio $p_1/s_1$ or $p_2/s_2$ or the compound product. The caption of Figure 2 seems to indicate that this is the transmittance ratio $p_2/s_2$. The authors should clarify this point and make a better link between words in the text and symbols in the equations]

**l.281/282**       …the difference of spectral radiance between the TANSO-FTS-2 and IASI with SNO condition. → …the difference of spectral radiances between TANSO-FTS-2 and IASI in SNO condition.

**l.282**   The variation range of brightness temperature between the TANSO-FTS-2 and IASI is wider than that of AIRS, then the SNO condition for IASI data is applied.

→ The range of brightness temperatures for the comparison between TANSO-FTS-2 and IASI is wider than that of AIRS, so the SNO condition for IASI also apply for AIRS.

[Is this what is meant? The initial sentence is not very clear]

**l.317**   …in the channels of $CH_4$. → …in the region around 7.6 µm covering the strong $CH_4$ signature.

[See below for the exact definition of "channel"]

**l.327**   …of the brightness temperatures difference… → …of the brightness temperature difference…

[plural not needed]

**l.328**   …for four channels…

[An exact definition of these channels is needed. How are the radiances in the wavenumber interval (to be specified) converted into scene brightness temperature? This is important since brightness temperature differences are then plotted and discussed]

**l.332**   …for the TANSO-FTS comparison. → …for comparison with the first TANSO-FTS instrument.

**l.337**  The comparison between the TANSO-FTS-2 and TANSO-FTS has a discrepancy in the low-temperature region, but we concluded that version v230231

→ The comparison between TANSO-FTS-2 and TANSO-FTS shows a significant difference for low-temperature scenes but we have to conclude that version v230231

**l.349**  …with TANSO-FTS-2 1° bin averaged along and cross-track angles.→ …with TANSO-FTS-2 in 1° bins of the pointing mirror angles along and across track.

[The authors could introduce the corresponding symbols. The symbol $\theta_{CT}$ has already been defined. Could one use $\theta_{AT}$? See a similar comment above]

**l.350**  The coincident observations between the TANSO-FTS-2 and the AIRS were selected +40° and -40° of the AIRS cross-track angle and the related along-track of the TANSO-FTS-2 listed in Table 1. For the IASI cross-track angle, +20° and -20° of the IASI data were selected.

[As such, the sentence in the text is hard to understand but Table 1 is helping. Again, using better notations for the angles will help both in the text and in the table. One could use $\theta_{CT}$(TANSO-FTS-2) and $\theta_{AT}$(TANSO-FTS-2) as well as $\theta_{CT}$(AIRS) and $\theta_{CT}$(IASI), insisting on the possibility of TANSO-FTS-2 to acquire footprints with large along-track angles as compared to the 3 other infrared sounders]

→ The coincident observations between TANSO-FTS-2 and AIRS in the 2O-SONO configuration presented in Figure 7 were selected with $\theta_{CT}$(TANSO-FTS-2) angles in the range +40° and -40° and $\theta_{CT}$(AIRS) angles in the range +40° and -40°, whereas the related $\theta_{CT}$(IASI) angles are in the range +20° and -20°as listed in Table 1.

**l.354**  …within the ±10° along-track angle.

→ …with $\theta_{AT}$(TANSO-FTS-2) angles in the range ±10°.

**l.354/355**  Figure 7(a) also … along-track angle.

[This sentence is redundant with the preceding one and with the proposed modification. It can be suppressed]

**l.356**  …of the TANSO-FTS-2→ …of TANSO-FTS-2

**l.357/358**  In contrast, … is not clear except for the $CH_4$ channel in a cross-track angle of 5° to 10°.

→ In contrast, … is not striking except for the $CH_4$ channel for $\theta_{CT}$(TANSO-FTS-2) in the range 5° to 10°.

**l.359** to **l.383**

[The reviewer recommends using symbols for $\theta_{CT}$(sounder) and $\theta_{AT}$(sounder) with sounder = TANSO-FTS-2, TANSO-FTS, AIRS and IASI to make the text more fluid, like what has been proposed above. The changes are not explicitly proposed here but should be made by the authors]

**l.366**  …and window. →…and in the atmospheric window region.

[Again the corresponding wavenumber interval should appear somewhere in the text or in a Table. See also the question on the conversion of spectral radiance (middle of the interval or integration over the full interval) to brightness temperature and then brightness temperature difference]

**l.371/372**  …results of comparing AIRS and IASI are not supported.

[This sentence is quite unclear]

**l.378** …to the temperature bias on TANSO-FTS… → …to the temperature bias in TANSO-FTS…

**l.380** the scene selection mirror… → the scene pointing mirror…

**l.384** and IASI is well. → and IASI is quite satisfactory.

**l.395** …for scenes brighter than… → …for scene temperatures brighter than…

**l.398/399** In the nominal temperature region, such as 280 K brightness temperature, the agreement… is well.

→ For scenes with brightness temperatures around 280 K, the agreement… is quite satisfactory.

**l.399** However, only TANSO-FTS suggests the brightness temperature bias in a cold scene over a high latitude region and is a challenge of TANSO-FTS.

→ However, comparisons of the 3 other infrared sounders with TANSO-FTS suggest a cold brightness temperature bias for cold scenes in high latitudes regions and this is an indication that the current products of this latter instrument have to be improved in these observation conditions.

**l.469** [The link in blue should be of the same font as the other ones used in other references]

**Equations**

More care should be given to the consistency between the equations and the notations defined either in the text or following them. As an example in (1), the variables $DAC_{scale b}$ and $DC_{offset b}$ are used whereas they appear as $DAC\_scale_b$ and $DC\_offset_b$ in the following text. The latter version is probably better to avoid subscript of subscript.

Similarly, $DN_b$ is used whereas $DN_{b,d}$ is appearing below. The corresponding definition could be "Digital count for each interferogram" rather than "Digital number for each interferogram".

In equation (5) the slowly varying terms $DC_b - a_{nlc,b}DC_b^2$ that appear in (4) have been neglected. This should be explained by saying that during the FFT numerical processing these terms are suppressed. It would have been interesting to plot a real interferogram to show how what is considered as the baseline is really constant or slowly varying. A comment about this would be welcome.

Equations (7) and (8) are quite difficult to follow without helping the reader to understand the various variables that should be explained upfront. In the Stokes vector $S_{T\_input}$, the symbol $B(T_{scene})$ is used. What is $B(T_{scene})$? This could be appropriate for a black body view, but not for an atmospheric scene. Or is it a "mean" scene temperature? [See question for **l.327**]

In **l.143**, 3 variables are introduced but two only seem to be defined just after their symbols.

The term $2p_1 (\sigma)q_1 (\sigma)$ should read $2p_1(\sigma)q_1(\sigma)$ [no space] in the matrices of **l.155** and **l.160**

The matrix $E$ in **l.160** is not defined explicitly and is (probably) the identity matrix. This should be made clear.

The following definitions should be explicit [superscripts are used since one needs this form to be consistent with similar symbols found in equations (10) to (13)]

$S^{ds}$ is the deep space signal [or radiance? This is to be defined]

$S^{obs}$ is the atmospheric signal [or radiance? This is to be defined]

$S^{ict}$ [written as $S_{bb}$ in (9)] is the signal [or radiance? This is to be defined] when viewing the calibration black body at temperature $T^{ict}$ [not $T_{bb}$ since $b$ is already used as subscript for band]

In **l.171**, one finds $B(T_{scene}) = L_{b,d}^{obs}$ [and this could be an answer to one of the above questions]. But why adding the additional subscript $d$ without further explanation? Is $d$ referring to one of the 4 "domains" or channels. The subscript $b$ seems to be used for the band (B4 or B5). It would be easier to just drop $d$ and explain that the equations pertain to both infrared bands and 4 domains with specific parameters for each of them. Is this why the wavenumber variable $\sigma$ is appearing in the variables $p1(\sigma)$, $q1(\sigma)$, $p2(\sigma)$ and $q2(\sigma)$?

In equation (10) the variables $B_{b,d}^{ict}$ [that could be simplified as $B_b^{ict}$ if the subscript $d$ is dropped] and $L_{b,d}^{m-obs}$ should be defined and/or related to previously defined variables.

In equations (15) to (20) the notation [$n$] is appearing without detailed explanations. This question has already been raised for **l.225**.

In equation (21) the variables A with superscript are dimensionless since their sum is unity. They are later defined as "view" and that is unclear. The symbol BS appears **l.233** and should be defined there as beam splitter. The same is true for SAA and OMA as already noted in comments for **l.229/230** and **l.231/232**.

Overall a more consistent use of symbols with subscripts/superscripts in the text and in the equations is needed.

**Tables**

**Table 1**

Title: Temporally and spatially co-incident conditions

→ Temporal and spatial coincidence conditions

First column header: Coincident type → Coincidence type

**Table 2**

Title: The averaged difference (Ave.) and deviation (SD.) of brightness temperatures between TANSO-FTS-2 and multi-satellite sensors with SNO

→ Average brightness temperature difference (mean) and standard deviation (stdv) between TANSO-FTS-2 and 4 other infrared sounders in the SNO configuration

[The proposed notations seem better than Ave. and SD.]

First column: [Since SNO is in the title, there is no need to repeat it after the name of the sounder. The column header could just be Sounder and this would help to avoid the unnecessary increased line spacing for TANSO-FTS. Try to reduce the width of the second column by replacing "Matchups" by "SNO". Include the line for sub-column headers defining mean and stdv into the line defining the channels. The word channel could even be replaced by the limits in wavenumber of the corresponding spectral domain]

**Table 3** [As above, the proposed title could be]

→ Average brightness temperature difference (mean) and standard deviation (stdv) between TANSO-FTS-2 and 4 other infrared sounders in the 2O-SONO configuration

[Try to have changes consistent with the ones proposed for Table 2]

**Figures**

**Figure 2**

Caption: p- and s-polarization against → p- and s-polarization $p_1(\sigma)/q_1(\sigma)$ against

**Figure 4**

Caption: for SNO (a) and SONO(b) → for SNO (a) and 2O-SONO (b)

**Figure 5**

…differences in 1 K gridded against window temperature… → …differences in 1 K bins against scene temperature…

[See below for the distinction between "window" and "channel"

…and each shade presents a standard deviation (1$\sigma$) for each 1 K grid

→ …and each shaded area presents the standard deviation (1$\sigma$) for each 1 K bin

**Figure 6**

Caption: average against window temperature → average against channel temperature

(d) window channels → (d) atmospheric window

[This is to avoid "window" used for channel (d). But as recommended above the limits of the domains or channels should be given]

**Figure 7**

… temperature difference in 1° grided bins average against …

→ … temperature difference in 1° angular bins against …

The shaded lines present the deviation (1$\sigma$) for each grid

→ The shaded area presents the standard deviation (1$\sigma$) for each 1° bin

[The definition "Coincident num." on the right vertical axis of each sub-panel (a), (c) and (e) is too close to the definition "Diff. Temp [K]" of the sub-panels (b), (d) and (f)]

**Figure 8**

The 1° × 1° gridded… → The 1° (AT) × 1° (CT) gridded…

**Figure 9**

… temperature difference in 1 K gridded average against …

→ … temperature difference in 1 K bins against …

…with deviation (shaded lines) → …with the corresponding standard deviation (shaded area)

(d) window channels → (d) atmospheric window

**Final recommendation**

The reviewer took from its time to propose a large number of modifications because he thinks that the work is important to pass the proper information to the wider user community of TANSO-FTS-2 and TANSO-FTS data (L1 and L2). Some minor changes are easy to implement, but a more challenging task will be for the authors to make a better link between the text, the notations and the equations. The reviewer is hoping that this can be done so that a revised version can reach the proper level of clarity for AMT readers. The huge amount of work done in comparing the radiometry of the Japanese sounders TANSO-FTS-2 and TANSO-FTS with IASI and AIRS is deserving it (if properly presented).

---

## Author Comment (AC1)

**Response to referee comments on manuscript amt-2022-129**

First of all, we would like to thank referee #1 for his/her constructive comments, which helped us to improve the manuscript. We replied all comments and questions as follows. The referee's comments are coped in blue text.

**Anonymous Referee #1**

General Comments:

**Referee:**

The authors report on the combination of two successful efforts: First, the calibration model was re-derived to account for nonlinear response, polarization sensitivity, and other features. Second, the spectral radiances were validated against three other instruments.

The new model is presented with 27 equations, which is both an advantage (very complete work) and a disadvantage (difficult to follow).

When possible, it would be beneficial to provide context for these reprocessed data products and validation activities by citing requirements, performance of other satellites, or accuracy thresholds linked to scientific goals. Also, the significance of the improved model could be highlighted by comparing accuracy & precision metrics between older and newer versions.

**Author's reply:**

Thank you very much for reviewing our manuscript. We revised our manuscript with changes tracked.

In the revised version, we added the additional explanation for the equations and symbols with consistency between the equations and text.

TANSO-FTS-2 and TANSO-FTS are unique instruments for observing both SWIR and TIR radiance spectra, simulatively. It can provide both the total and partial column concertation of GHG. In other words, they can provide the near surface (0 to around 4km above from ground) GHG concentration, globally. Those products have advantage for understanding the global carbon cycle. To retrieve the accurate partial column concentration, the angle depended or scene brightness temperature depended bias in radiance spectral domain is undesirable. As for TIR sounder, the radiometric and spectral consistency among other sounders is important. Then, we developed new estimation method for 2-orthogonal simultaneous off-nadir overpass, updated the calibration procedure for spectral radiance and improved the spectral quality in off-nadir observations.

To clarify for this science objective, we add the following sentences in the section 1.

"Simultaneous spectral radiance observation for SWIR and TIR supports retrieving new partial column concentration of $CO_2$ and $CH_4$ as well as the total column concentration which are conventional products. The partial column concentration has sensitivities for the near surface (ground to around 4 km altitude) and upper troposphere (between 4 and around 12 km altitude) of $CO_2$ and $CH_4$ concentrations. These products lead to new applications for local emission estimation (Kuze et al., 2022). "

"To provide the radiometric and spectral consistency among the TIR sounders as well as the accurate partial column concentration, the angle dependent or scene radiance dependent bias in radiance spectral domain is undesirable. Then, we showed that the spectral radiance for TANSO-FTS-2 TIR bands is consistent with the intercalibration data of the other TIR sounders mentioned above, with time-series, wavenumber, and the incident angle dependencies."

As for other satellite performance, we added the overview of characterization results for other sensors are implemented in session 4 with additional references.

The key messages are as follows;

"Aumann et al. 2019 have studied the long-term stability of AIRS spectra as compared with calculated spectra over Tropical Ocean at night and found that the trend of all AIRS longwave channels in the surface sensitive channels was quite small (2 mK/yr). In addition, AIRS and IASI are well characterized and the bias of these sensors are reported less than 0.2 K (Jouglet et al., 2014). Then, our calibration target is to provide the consistent spectral radiance among the TIR sounder for full coverage of TANSO-FTS-2 observation angless."

Also, the comparison results between previous (old) and new products are highlighted in table 2 and section 4.1.

Specific Comments and Technical Corrections:

**Referee:**
Finally, the authors are encouraged to make a number of minor English corrections:

Line 12: 0.2cm$^{-1}$ -> separate number & unit

**Author's reply:**
We corrected the word "0.2cm$^{-1}$" to "0.2 cm$^{-1}$" in the revised manuscript.

**Author's reply:**

We changed a small letter of "earth" to capitalized "Earth".

**Referee:**
Line 45: Characterization of these spectral radiance is essential (reword)

**Author's reply:**

We modified the word "Characterization of these spectral radiance is essential" to "The calibrated spectral radiance is essential"

**Referee:**
Line 204: contamination -> combination

**Author's reply:**

We changed the word "contamination" to "combination" in the revised manuscript.

**Referee:**
Line 311 & 579: grided -> gridded

**Author's reply:**

We corrected the word "grided" to "gridded" in line 311 and to "angular bin" in line 579 in the revised manuscript, respectively.

**Referee:**
Line 387: Table.3.

**Author's reply:**

We corrected the word "Table.3" to "Table 3" in the revised manuscript.

**Referee:**
Line 394: orthogoanl -> orthogonal

**Author's reply:**

We corrected the word "orthogoanl" to "orthogonal" in the revised manuscript.

**Referee:**
Line 401: TANO -> TANSO

**Author's reply:**

We corrected the word "TANO" to "TANSO" in the revised manuscript.

**Author's reply:**

We removed the redundant word "bias" in the revised manuscript.

**Author's reply:**

We corrected the word "vicairous" to "vicarious" in the revised manuscript.

**Author's reply:**

We corrected the word "londitude" to "longitude" in Fig 4 in the revised manuscript.

End of document

---

## Author Comment (AC2)

**Response to referee comments on manuscript amt-2022-129**

First of all, we would like to thank the referee (Dr. Claude Camy-Peyret) for his constructive comments, which helped to improve the manuscript. We replied all comments and questions as follows. The referee's comments are coped in blue text.

General Comments

Overall assessment

**Referee:**

This is an important paper from the Japanese team of GOSAT (with one US co-author) on the radiometric calibration of TANSO-FTS-2 in the TIR bands with some discussion of the earlier TANSO- FTS instrument. The paper is rather technical but deserves publishing in AMT after accounting for the modifications suggested in this review and consideration of some questions. Figures, although numerous, are presenting in a compact and visual manner the main results of the radiometric differences between the Japanese instruments and two well validated infrared sounders (IASI from CNES and AIRS from JPL). An extensive reference is done to a previous paper on the subject by Suto et al. (2021). But as suggested below, the authors should help the reader to read this manuscript in a more stand-alone mode. A final recommendation is given at the end of this review.

Note that remarks from an anonymous reader at https://doi.org/10.5194/amt-2022-129-RC1 are endorsed by the present review and should also be taken into account.

**Author's reply:**

Thank you very much for reviewing our manuscript. We revised our manuscript with change tracks.

In the revised version, we respond all comments as well as the comments from an anonymous reader at https://doi.org/10.5194/amt-2022-129-RC1.

Suggested detailed modifications:

**Referee:**

The reviewer is using **l.xxx** (in bold) for the line to be modified and the convention old text → new text for the proposed changes. [A text between brackets is a comment/question/explanation as this is done here!]

General comment: one could just use TANSO-FTS-2 and TANSO-FTS instead of "the TANSO-FTS-2" and "the TANSO-FTS". This has been done in the following proposed modifications but not everywhere in the text.

**Author's reply:**

We corrected the word of "the TANSO-FTS" and "the TANSO-FTS-2" to "TANSO-FTS" and "TANSO-FTS-2" in the revised manuscript, respectively.

**Author's reply:**

We corrected the phrase of "… to longwave Thermal InfraRed radiation (TIR) with 0.2cm$^{-1}$ spectral intervals." to "… to the longwave Thermal InfraRed radiation (TIR) with 0.2 cm$^{-1}$ spectral sampling and the corresponded spectral resolution (Full width at half maximum: FWHM) of TIR region is less than 0.27 cm$^{-1}$."

**Author's reply:**

We corrected the manuscripts as follows;

"… consistent with the intercalibration data of the other TIR sounders mentioned above, with time-series, wavenumber, and the incident angle dependencies."

**Author's reply:**

We corrected the phrase of "… a non-linear response…" to "… the non-linear response of the infrared detectors …".

**Referee:**

**l.99** ...considers up to the linear and quadratic terms.

→ ...considers only the linear and quadratic terms (neglecting the cubic one).

**Author's reply:**

We corrected the phrase of "… considers up to the linear and quadratic terms" to "… considers only the linear and quadratic terms (neglecting the cubic one)".

**Referee:**

**l.105** with a quadratic term...

→ with a quadratic term only...

**Author's reply:**

We added the word of "only" in this phrase.

**Referee:**

**L.140/l.142** Choose between Mueller or Muller but do not use both forms! What about Müller?]

**Author's reply:**

We replaced "Mueller" with "Müller" in the revised manuscript.

**Referee:**

**l.142** ...and CT rotation...

→ ...and cross-track (CT) rotation angle (called $\theta_{CT}$ in the following) ...

[In several instances specific notations for the various parameters or variables are used in the equations but not defined early enough for the reader to follow them smoothly. The reviewer has tried to improve this situation. But the question of equations is also discussed below in a dedicated section of this review]

**Author's reply:**

We corrected the phrase of "… and CT rotation …" to "…and cross-track (CT) rotation angle (called $\theta_{CT}$ in the following)".

We revised the notations for the various parameters or variables in the revised manuscript.

**Referee:**

**l.176** The term in equation (11) already corrected the non-linear effects.

→ The multiplicative factor of the first term of equation (10) is called $Cal_b$ in the following equation (11) and includes the non-linearity correction. [Check if it is OK and see the discussion below on the equations concerning the subscripts $b$ and $d$]

**Author's reply:**

We modified the equation (11) and related sentence. The subscript d means "direction of FTS scan-motion". In the revised manuscript, we removed it to avoid the complexity for reader.

Also, almost all parameters are depended on wavenumber. Then, we mentioned it in the text and simplified notations in equations.

**Referee:**

**l.179** Equation (11) should read $Cal_b$ = [...]

**Author's reply:**

We modified the equation (11) with $Cal_b$.

**Referee:**

**l.181** ...equation (6), the equation (11) is extracted as equation (12).

→ ...equation (6), equation (11) can be recast as equation (12).

**Author's reply:**

We corrected the phrase "… equation (6), the equation (11) is extracted as equation (12)" to "… equation (6), equation (11) can be recast as equation (12)".

**Referee:**

**l.186** [The symbol $a_2$ is not specifically defined in equation (12). It should (perhaps) be related to $a_{nlc,b}$ of equation (2). The variable $p_g$ is not precisely defined. Overall the text there is hard to follow!]

[Check if equation (12) coul read:] $C_{alb}$ = [...] = [...]

**Author's reply:**

We corrected the word of "$a_2$" to" $a\_nlc_b$". The notation of $a_2$ is non-linearity coefficient for each band. As we described in previous response, we improved the text, notations and equations in the revised manuscript.

**Referee:**

**l.189** the polarization axis of the internal optics is rotated at 90° from the nadir observation.

[This is not quite clear and the reviewer is suggesting the following]

→ the pointing mirror is rotated along its axis by ±90° (from the nadir observation) to view the deep space (ds) or the internal calibration target (ict).

[This would be consistent and will recall the notations introduced earlier in the text. The polarization effect is implicit because the incidence angle on the mirror would change. Relating the angle to the CT angle $\theta_{CT}$ already introduced would be even better. Could one specify that $\theta_{CT} = 0$ is exact nadir?]

**Author's reply:**

We modified the phrase of "the polarization axis of the internal optics is rotated at 90° from the nadir observation" to "the pointing mirror is rotated along its axis by +/- 90° (from $\theta_{CT} =0$, exact nadir observation) to view the deep space or the black body calibration target".

**Referee:**

**l.190** ...the difference in input optical angles.

→ ...the difference in incidence angle on the pointing mirror.

**Author's reply:**

We corrected the phrase of "… the difference in input optical angles" as suggested by reviewer.

**Referee:**

**l.202** ...(ict or internal target) is a contamination of a direct emission from the blackbody and reflected...

→ ...(ict for internal target in the various symbols used) is a combination of the direct emission from the black body (at the temperature $T_{ict}$) and the reflected...

[One has to decide between black body and blackbody consistently in the text]

**Author's reply:**

We modified the symbols for black body as bb, instead of ict, and use the phrase of "black body" in the revised manuscript.

**Referee:**

**l.223** for p and s

→ for p and s polarizations (see subscripts p and s in the corresponding symbols)

**Author's reply:**

We corrected the word and the corresponding symbols.

**Referee:**

**l.224** Transmittance for p- and s-polarization signals for internal optics

→ Transmittance for the p- and s-polarized beams within the FTS

**Author's reply:**

We corrected the phrase as "Transmittance for the p- and s-polarized beams within the FTS including after optics."

In our case, both FTS module and after optics have to be considered for its polarization sensitivity. Then, we also mentioned that FTS including after optics.

**Referee:**

**l.225** Radiance for temperature $T_{ict}$, and wavenumber $\sigma_b[n]$

→ Radiance for black body at a temperature $T_{ict}$ and wavenumber $\sigma_b[n]$

[The authors should better explain what is $[n]$. Is it the channel index ($CO_2$, window, $O_3$, $CH_4$) used in Tables 1, 2 and 3 as well as in the figures?]

**Author's reply:**

We corrected the phrase. The indication of [n] is redundant. It means the index of each band. It not related the specific spectral ranges. Then, we removed this indicator in the revised manuscript to avoid the complexity. Generally, the radiance for black body at temperature T is depended on wavenumber. In the text, we added the phrase for this. Also, we simplified the description of equations in the revised manuscript.

**Referee:**
**l.226** ...between calibration angles (ICT and Deep-space)
→ ...between different pointing mirror angles towards the black body (ict in the various symbols) or deep space (ds in the various symbols)

**Author's reply:**
We corrected the phrase, and use the bb as the black body symbol.

**Referee:**

**l.229/230** [The symbol SAA is not precisely defined and could induce confusion with the South Atlantic Anomaly. Could the authors propose a different acronym?]

**Author's reply:**

In our case, SAA means Scene Selection Assembly, which is the mounting structure for the pointing mechanism. As suggested by reviewer, the reader will confuse this phrase. Then, we use other phrase as Pointing Mechanism Assembly (PMA) instead of Scene Selection Assembly (SSA).

**Referee:**

**l.231/232** [Same question for the definition of OMA]

**Author's reply:**

OMA is Optics Mount Assembly. To make direct link with housekeeping telemetry which provided in the product, we corrected the phrase and notation in the revised manuscript as Integrated Optics Assembly (IOA) instead of OMA.

**Referee:**

**l.238** ...complex index of the mirror sample...

→ ...complex index of refraction of the mirror material (with coating)....

**Author's reply:**

We corrected the phrase as suggested by reviewer.

**Referee:**

**l.239** ...actual mirror.

→ ...actual flight mirror.

**Author's reply:**

We corrected the phrase as suggested by reviewer.

**Referee:**

**l.240** [One could add:]
→ A star as superscript is used for the complex conjugate in equations (25) and (26).

**Author's reply:**

We added the sentence of "A star as superscript is used for the complex conjugate in equations (25) and (26)."

**Referee:**

**l.241** Equation (22)

[The authors should define more precisely CTang and ATang and make the link with $\theta_{CT}$ already introduced or could possibly use $\theta_{AT}$? The angles CTang and ATang are appearing in Figure 3 and are called "AT angle" and "CT angle". This is fine but this is introducing new notations. TANSO-FTS is on the same footing as IASI and AIRS and the corresponding viewing direction across track is called "scan angle". A better explanation of these various angles (in particular a more consistent choice of notations) would be useful. The same type of comment is appearing several times below]

**Author's reply:**

We corrected the caption of figure 3 as well as the text. We introduced $\theta_{CT}$ and $\theta_{AT}$ as suggested by reviewer in the revised manuscript.

**Referee:**
**L.259** [Is there a difference between the "pointing mirror" and the "scan mirror"? The scan mirror could be understood as the moving mirror of the interferometer. This is to be clarified and consistent notations should be used in equations (27) and in **l.234**]

**Author's reply:**
The phrase of "pointing mirror" and "scan mirror" is same meaning. To avoid the confusion, we only used the "pointing mirror" in the revised manuscript.
In this manuscript, we don't mention the moving mirror of the interferometer.

**Referee:**
**l.272** ...the ratio of the p and s internal optics against...
→...the ratio of the p and s transmission against...
[It is unclear at this point if it is the ratio of the signals at the output of the detector or if it is the ratio p1/s1 or p2/s2 or the compound product. The caption of Figure 2 seems to indicate that this is the transmittance ratio p2/s2. The authors should clarify this point and make a better link between words in the text and symbols in the equations]

**Author's reply:**
The figure 2 present the $p_2^2/q_2^2$ on vertical axis. We corrected the label of figure 2 and made better link between text and figure.

**Referee:**

**l.281/282** ...the difference of spectral radiance between the TANSO-FTS-2 and IASI with SNO condition.

→ ...the difference of spectral radiances between TANSO-FTS-2 and IASI in SNO condition.

**Author's reply:**
We corrected the phrase as suggested by reviewer.

**Referee:**
**l.282** The variation range of brightness temperature between the TANSO-FTS-2 and IASI is wider than that of AIRS, then the SNO condition for IASI data is applied.

→ The range of brightness temperatures for the comparison between TANSO-FTS-2 and IASI is wider than that of AIRS, so the SNO condition for IASI also apply for AIRS.

**Author's reply:**
We corrected the sentence as suggested by reviewer.

**Referee:**
**l.317** ...in the channels of CH4.

→ ...in the region around 7.6 μm covering the strong $CH_4$ signature. [See below for the exact definition of "channel"]

**Author's reply:**
We added the following sentence for clear definition of spectral range and the calculation process for comparing the brightness temperature between TANSO-FTS-2 and other sounders in the revised manuscript.

"We focused on the comparison in the following spectral ranges: $CO_2$ spectral range (681.99 - 691.66 $cm^{-1}$), atmospheric window channel (900.3 - 903.78 $cm^{-1}$), $O_3$ spectral range (1030.08 - 1039.69 $cm^{-1}$), and $CH_4$ spectral range (1304.36 - 1306.68 $cm^{-1}$) same as previous our estimation (Suto et al., 2021). Since the spectral resolution of AIRS and IASI is different from that of TANSO-FTS-2, we convolve the TANSO-FTS-2 spectra with AIRS spectral response function to comparing these data. After that, the average brightness temperature for four spectral regions is computed for both sounders. The same convolution and averaging processes are also applied to IASI data. "
Also, we corrected the sentence as suggested by reviewer.

**Referee:**
**l.327** ...of the brightness temperatures difference...

→ ...of the brightness temperature difference... [plural not needed]

**Author's reply:**

We corrected the plural of brightness temperature.

**Referee:**
**l.328** ...for four channels...
[An exact definition of these channels is needed. How are the radiances in the wavenumber interval (to be specified) converted into scene brightness temperature? This is important since brightness temperature differences are then plotted and discussed]

**Author's reply:**
We added the definition of four spectral ranges and processing scheme as response of previous comments.

**Referee:**
**l.332** ...for the TANSO-FTS comparison.
→ ...for comparison with the first TANSO-FTS instrument.

**Author's reply:**
We corrected the phrase of "… for the TANSO-FTS comparison" to "… for comparison with the first TANSO-FTS instrument."

**Referee:**
**l.337** The comparison between the TANSO-FTS-2 and TANSO-FTS has a discrepancy in the low- temperature region, but we concluded that version v230231
→ The comparison between TANSO-FTS-2 and TANSO-FTS shows a significant difference for low- temperature scenes but we have to conclude that version v230231

**Author's reply:**
We corrected the phrase as suggested by reviewer.

**Referee:**
**l.349** ...with TANSO-FTS-2 1° bin averaged along and cross-track angles.
→ ...with TANSO-FTS-2 in 1° bins of the pointing mirror angles along and across track.
[The authors could introduce the corresponding symbols. The symbol $\theta_{CT}$ has already been defined. Could one use $\theta_{AT}$? See a similar comment above]

**Author's reply:**
We corrected the phrase as suggested by reviewer. We also use $\theta_{AT}$ for along-track angle.

**Referee:**
**l.350** The coincident observations between the TANSO-FTS-2 and the AIRS were selected +40° and -40° of the AIRS cross-track angle and the related along-track of the

TANSO-FTS-2 listed in Table 1. For the IASI cross-track angle, +20° and -20° of the IASI data were selected.
[As such, the sentence in the text is hard to understand but Table 1 is helping. Again, using better notations for the angles will help both in the text and in the table. One could use $\theta_{CT}$(TANSO-FTS-2) and $\theta_{AT}$(TANSO-FTS-2) as well as $\theta_{CT}$(AIRS) and $\theta_{CT}$(IASI), insisting on the possibility of TANSO-FTS-2 to acquire footprints with large along-track angles as compared to the 3 other infrared sounders]

→  The coincident observations between TANSO-FTS-2 and AIRS in the 2O-SONO configuration presented in Figure 7 were selected with $\theta_{CT}$(TANSO-FTS-2) angles in the range +40° and -40° and $\theta_{CT}$(AIRS) angles in the range +40° and -40°, whereas the related $\theta_{CT}$(IASI) angles are in the range +20° and -20°as listed in Table 1.

**Author's reply:**
We corrected the sentence as suggested by reviewer.

**Referee:**
**l.354** ...within the ±10° along-track angle.
→  ...with $\theta_{AT}$(TANSO-FTS-2) angles in the range ±10°.

**Author's reply:**
We corrected the phrase as suggested by reviewer.

**Referee:**
**l.354/355** Figure 7(a) also ... along-track angle.
[This sentence is redundant with the preceding one and with the proposed modification. It can be suppressed]

**Author's reply:**
We removed this redundant sentence in the revised manuscript.

**Referee:**
**l.356** ...of the TANSO-FTS-2
→  ...of TANSO-FTS-2

**Author's reply:**
We corrected the word of "TANSO-FTS-2".

**Referee:**
**l.357/358** In contrast, ... is not clear except for the $CH_4$ channel in a cross-track angle of 5° to 10°.
→  In contrast, ... is not striking except for the $CH_4$ channel for $\theta_{CT}$(TANSO-FTS-2) in the range 5° to 10°.

**Author's reply:**
We corrected the phrase as suggested by reviewer.

[The reviewer recommends using symbols for $\theta_{CT}$(sounder) and $\theta_{AT}$(sounder) with sounder = TANSO- FTS-2, TANSO-FTS, AIRS and IASI to make the text more fluid, like what has been proposed above. The changes are not explicitly proposed here but should be made by the authors]

**Author's reply:**
We modified the phrase as suggested by reviewer.

$\rightarrow$ ...and in the atmospheric window region.
[Again the corresponding wavenumber interval should appear somewhere in the text or in a Table. See also the question on the conversion of spectral radiance (middle of the interval or integration over the full interval) to brightness temperature and then brightness temperature difference]

**Author's reply:**
We modified the phrase and describe the specific spectral ranges and processing scheme in the text.

[This sentence is quite unclear]

**Author's reply:**
We modified the phrase as follows;
", even though the comparison between TANFO-FTS-2 and AIRS, between TANSO-FTS-2 and IASI are not indicated a cross-track angle dependence."

$\rightarrow$ ...to the temperature bias in TANSO-FTS...

**Author's reply:**
We corrected the phrase as suggested by reviewer.

$\rightarrow$ the scene pointing mirror...

**Author's reply:**
We corrected the phrase as "this difference may indicate a pointing angle dependence of the pointing mirror, ".

**Referee:**
**l.384** and IASI is well.
→ and IASI is quite satisfactory.

**Author's reply:**
We corrected the phrase as suggested by reviewer.

**Referee:**
**l.395** ...for scenes brighter than...
→ ...for scene temperatures brighter than...

**Author's reply:**
We corrected the phrase as suggested by reviewer.

**Referee:**
**l.398/399** In the nominal temperature region, such as 280 K brightness temperature, the agreement... is well.
→ For scenes with brightness temperatures around 280 K, the agreement... is quite satisfactory.

**Author's reply:**
We corrected the phrase as suggested by reviewer.

**Referee:**
**l.399** However, only TANSO-FTS suggests the brightness temperature bias in a cold scene over a high latitude region and is a challenge of TANSO-FTS.
→ However, comparisons of the 3 other infrared sounders with TANSO-FTS suggest a cold brightness temperature bias for cold scenes in high latitudes regions and this is an indication that the current products of this latter instrument have to be improved in these observation conditions.

**Author's reply:**
We corrected the sentence as suggested by reviewer.

**Referee:**
**l.469** [The link in blue should be of the same font as the other ones used in other references]

**Author's reply:**
We corrected the font and link in the revised manuscript.

**Referee:**
**Equations**
More care should be given to the consistency between the equations and the notations defined either in the text or following them. As an example in (1), the variables $DAC\_scale_b$ and $DC\_offset_b$ are used whereas they appear as $DAC\_scale_b$ and $DC\_offset_b$ in the following text. The latter version is probably better to avoid subscript of subscript.

**Author's reply:**
We corrected the notation and subscript as suggested by reviewer.

Similarly, $DN_b$ is used whereas $DN_{b,d}$ is appearing below. The corresponding definition could be "Digital count for each interferogram" rather than "Digital number for each interferogram".

**Author's reply:**
We corrected the definition of DN as suggested by reviewer.

In equation (5) the slowly varying terms $DC_b - a_{nlc,b}DC_b^2$ that appear in (4) have been neglected. This should be explained by saying that during the FFT numerical processing these terms are suppressed. It would have been interesting to plot a real interferogram to show how what is considered as the baseline is really constant or slowly varying. A comment about this would be welcome.

**Author's reply:**
We added the sentence of "During the fast-Fourier transform numerical processing, the term of $(DC_b - a\_nlc_b DC_b^2)$ are suppressed."

For the actual interferogram of TANSO-FTS-2, the baseline is slowly varying. In contrast, TANSO-FTS has constant baseline. The double pendulum type of FTS mechanism is accommodated both TANSO-FTS and TANSO-FTS-2. However, the electronic chains are different. The electronics for TANSO-FTS is implemented the high path filter for the interferogram signal. Then, the small variation is removed via this electronic filter. In contrast, the electronics for TANSO-FTS-2 can transmit low frequency component of interferogram signal. Then, it appears on the interferogram as the slowly varying signal.
However, these frequencies of small variations are much smaller than that of in-band signal. Then, it is also negligible.

Equations (7) and (8) are quite difficult to follow without helping the reader to understand the various variables that should be explained upfront. In the Stokes vector $S_{T\_input}$, the symbol $B(T_{scene})$ is used. What is $B(T_{scene})$? This could be appropriate for a black body view, but not for an atmospheric scene. Or is it a "mean" scene temperature? [See question for **l.327**]

**Author's reply:**
We revised the notation and add the additional explanation for the equation in the revised manuscript.

In **l.143**, 3 variables are introduced but two only seem to be defined just after their symbols. The term *2p1 (σ)q1 (σ)* should read *2p1(σ)q1(σ)* [no space] in the matrices of **l.155** and **l.160**

**Author's reply:**
We corrected the phrase, and also modified the lines for the explanation of symbols.

The matrix $E$ in **l.160** is not defined explicitly and is (probably) the identity matrix. This should be made clear.

**Author's reply:**
We add the explanation of E matrix in the text.

The following definitions should be explicit [superscripts are used since one needs this form to be consistent with similar symbols found in equations (10) to (13)]

$S^{ds}$ is the deep space signal [or radiance? This is to be defined]

$S^{obs}$ is the atmospheric signal [or radiance? This is to be defined]

$S^{ict}$ [written as $S_{bb}$ in (9)] is the signal [or radiance? This is to be defined] when viewing the calibration black body at temperature $T^{ict}$ [not $T_{bb}$ since $b$ is already used as subscript for band]

**Author's reply:**
We add the explanations for missing symbols in the revised manuscript.

In **l.171**, one finds $B(T_{scene}) = L_{b,d}^{obs}$ [and this could be an answer to one of the above questions]. But why adding the additional subscript $d$ without further explanation? Is $d$ referring to one of the 4 "domains" or channels. The subscript $b$ seems to be used for the band (B4 or B5). It would be easier to just drop $d$ and explain that the equations pertain to both infrared bands and 4 domains with specific parameters for each of them. Is this why the wavenumber variable σ is appearing in the variables $p1(σ), q1(σ), p2(σ)$ and $q2(σ)$?

**Author's reply:**
We corrected symbols and subscript in the equation. In addition, we add the clear definitions for symbols in the revised manuscript.

In equation (10) the variables $B_{b,d}^{ict}$ [that could be simplified as $B_b^{ict}$ if the subscript $d$ is dropped] and $L_{b,d}^{m\_obs}$ should be defined and/or related to previously defined variables.

**Author's reply:**
We add the definitions for variables in the revised manuscript.

In equations (15) to (20) the notation [*n*] is appearing without detailed explanations. This question has already been raised for **l.225**.

**Author's reply:**
It means the index of each channel. However, it is not important for reader. We removed the notation [n] to avoid the confusion by reader and simplify the description.

In equation (21) the variables A with superscript are dimensionless since their sum is unity. They are later defined as "view" and that is unclear. The symbol BS appears **l.233** and should be defined there as beam splitter. The same is true for SAA and OMA as already noted in comments for **l.229/230** and **l.231/232**.

**Author's reply:**
We added the definition of "$A^{xx}$" and corrected consistently the notation and symbols in the revised manuscript.

Overall a more consistent use of symbols with subscripts/superscripts in the text and in the equations is needed.

**Author's reply:**
We carefully corrected the symbols with subscripts in the text and in the equations.

**Referee:**
**Table 1**
Title: Temporally and spatially co-incident conditions
→ Temporal and spatial coincidence conditions
First column header: Coincident type
→ Coincidence type

**Author's reply:**
We corrected the word in the revised manuscript.

**Referee:**
**Table 2**
Title: The averaged difference (Ave.) and deviation (SD.) of brightness temperatures between TANSO- FTS-2 and multi-satellite sensors with SNO
→ Average brightness temperature difference (mean) and standard deviation (stdv) between TANSO-FTS-2 and 4 other infrared sounders in the SNO configuration
[The proposed notations seem better than Ave. and SD.]

**Author's reply:**
We corrected notations for average and standard deviation as suggested by reviewer.

First column: [Since SNO is in the title, there is no need to repeat it after the name of the sounder. The column header could just be Sounder and this would help to avoid the unnecessary increased line spacing for TANSO-FTS. Try to reduce the width of the second column by replacing "Matchups" by "SNO". Include the line for sub-column headers defining mean and stdv into the line defining the channels. The word channel could even be replaced by the limits in wavenumber of the corresponding spectral domain]

**Author's reply:**
We corrected the word as suggested by reviewer.

**Referee:**
**Table 3** [As above, the proposed title could be]
→ Average brightness temperature difference (mean) and standard deviation (stdv) between TANSO-FTS-2 and 4 other infrared sounders in the 2O-SONO configuration [Try to have changes consistent with the ones proposed for Table 2]

**Author's reply:**
We corrected the word as suggested by reviewer. We also corrected table 2.

**Referee:**
**Figure 2**
Caption: p- and s-polarization against
→ p- and s-polarization $p1(\sigma)/q1(\sigma)$ against

**Author's reply:**
We corrected the caption and label of figure 2.

**Referee:**
**Figure 4**
Caption: for SNO (a) and SONO(b)
→ for SNO (a) and 2O-SONO (b)

**Author's reply:**
We corrected the caption as suggested by reviewer.

**Referee:**
**Figure 5**
...differences in 1 K gridded against window temperature...
→...differences in 1 K bins against scene temperature...
[See below for the distinction between "window" and "channel"

...and each shade presents a standard deviation (1σ) for each 1 K grid
→...and each shaded area presents the standard deviation (1σ) for each 1 K bin

**Author's reply:**
We corrected the phrase as suggested by reviewer.
We added the definition of "spectral range" in the revised manuscript. In addition, we named four analysis spectral range as $CO_2$ spectral range, $CH_4$ spectral range, $O_3$ spectral range, and atmospheric window channel with processing scheme.

**Referee:**
**Figure 6**
Caption: average against window temperature
→  average against channel temperature
(d) window channels
→(d) atmospheric window
[This is to avoid "window" used for channel (d). But as recommended above the limits of the domains or channels should be given]

**Author's reply:**
We add the definition of spectral range for clear understanding by reader as response in previous comment.

**Referee:**
**Figure 7**
... temperature difference in 1° grided bins average against ...
→... temperature difference in 1° angular bins against ...

The shaded lines present the deviation (1σ) for each grid
→  The shaded area presents the standard deviation (1σ) for each 1° bin
[The definition "Coincident num." on the right vertical axis of each sub-panel (a), (c) and (e) is too close to the definition "Diff. Temp [K]" of the sub-panels (b), (d) and (f)]

**Author's reply:**
We corrected the phrase and figure label locations as suggested by reviewer.

**Referee:**
**Figure 8**
The 1° × 1° gridded...
→  The 1° (AT) × 1° (CT) gridded...

**Author's reply:**
We corrected the phrase as suggested by reviewer.

**Referee:**
**Figure 9**

... temperature difference in 1 K gridded average against ...
→... temperature difference in 1 K bins against ...

...with deviation (shaded lines)
→...with the corresponding standard deviation (shaded area)

 (d) window channels
→(d) atmospheric window

**Author's reply:**
We corrected the phrase as suggested by reviewer.

**Referee:**
**Final recommendation**
The reviewer took from its time to propose a large number of modifications because he thinks that the work is important to pass the proper information to the wider user community of TANSO-FTS-2 and TANSO-FTS data (L1 and L2). Some minor changes are easy to implement, but a more challenging task will be for the authors to make a better link between the text, the notations and the equations. The reviewer is hoping that this can be done so that a revised version can reach the proper level of clarity for AMT readers. The huge amount of work done in comparing the radiometry of the Japanese sounders TANSO-FTS-2 and TANSO-FTS with IASI and AIRS is deserving it (if properly presented).

**Author's reply:**
We corrected, added and modified our manuscript based on the reviewer's suggestion. These reviewer's comments and questions are very helpful for improve our manuscript.

End of document